# The Lipid–Heart Hypothesis and the Keys Equation Defined the Dietary Guidelines but Ignored the Impact of *Trans*-Fat and High Linoleic Acid Consumption

**DOI:** 10.3390/nu16101447

**Published:** 2024-05-11

**Authors:** Mary T. Newport, Fabian M. Dayrit

**Affiliations:** 1Independent Researcher, Spring Hill, FL 34610, USA; 2Department of Chemistry, Ateneo de Manila University, Loyola Heights, Quezon City 1108, Philippines

**Keywords:** lipid–heart hypothesis, Ancel Keys, saturated fat, *trans*-fat, polyunsaturated fat, cholesterol, heart disease, dietary guidelines

## Abstract

In response to a perceived epidemic of coronary heart disease, Ancel Keys introduced the lipid–heart hypothesis in 1953 which asserted that high intakes of total fat, saturated fat, and cholesterol lead to atherosclerosis and that consuming less fat and cholesterol, and replacing saturated fat with polyunsaturated fat, would reduce serum cholesterol and consequently the risk of heart disease. Keys proposed an equation that would predict the concentration of serum cholesterol (ΔChol.) from the consumption of saturated fat (ΔS), polyunsaturated fat (ΔP), and cholesterol (ΔZ): ΔChol. = 1.2(2ΔS − ΔP) + 1.5ΔZ. However, the Keys equation conflated natural saturated fat and industrial *trans*-fat into a single parameter and considered only linoleic acid as the polyunsaturated fat. This ignored the widespread consumption of *trans*-fat and its effects on serum cholesterol and promoted an imbalance of omega-6 to omega-3 fatty acids in the diet. Numerous observational, epidemiological, interventional, and autopsy studies have failed to validate the Keys equation and the lipid–heart hypothesis. Nevertheless, these have been the cornerstone of national and international dietary guidelines which have focused disproportionately on heart disease and much less so on cancer and metabolic disorders, which have steadily increased since the adoption of this hypothesis.

## 1. Introduction

One of the most common warnings in dietary recommendations is to avoid saturated fat and to replace saturated fat with polyunsaturated fat. The 1980 *Dietary Guidelines for Americans* (*DGA*) warned the following: “Avoid Too Much Fat, Saturated Fat, and Cholesterol” [1]. Although the warning against dietary cholesterol was revised in 2015, the warning has remained to limit saturated fat to 10% of energy but without guidance for limits on polyunsaturated fat. In 1961, the American Heart Association (AHA) defined saturated fat as “the fat in whole milk, cream, butter, cheese and meat” [2], and this definition has persisted. The *DGA 2020* states the following: “Saturated fat is commonly found in higher amounts in high-fat meat, full-fat dairy products (e.g., whole milk, ice cream, cheese), butter, coconut oil, and palm kernel and palm oil” ([3], p. 44). However, this is a misleading description of “saturated fat” because it does not mention *trans*-fat products, such as margarine and shortening, which were historically conflated with natural saturated fat. Further, the *DGA* promoted the consumption of linoleic acid without limits. Recent reviews have addressed the lack of evidence that saturated fat in general or in specific foods, such as milk and eggs, causes cardiovascular disease (CVD) or that reducing saturated fat intake lowers CVD risk, the importance of LDL-C particle size and distribution pattern in CVD rather than level of total LDL-C, and the importance of the food matrix and overall dietary pattern which affect digestion, absorption, and other properties of specific nutrients [4,5,6]. Other critiques have focused on exposing reliance on insufficient evidence [7], food industry pressure [8], and biases and conflicts of interest [9,10] in the formulation of dietary guidelines. This historical review will trace the evolution of this hypothesis, its role in the development of dietary guidelines, and its failure to differentiate natural sources of saturated fat from industrial *trans*-fats and to place limits on polyunsaturated fat.

## 2. The Ancel Keys Equations and the Diet–Lipid Hypothesis

In the 1950s, Ancel Keys, in his effort to understand the reason for the perceived increase in the incidence of heart disease in the US population, conducted brief feeding studies of usually 4 weeks in small groups of mostly institutionalized subjects with mixed results. Despite consuming the same strictly controlled meals, Keys found significant intra-individual differences from week to week of 10–12% and marked inter-individual differences in total serum cholesterol (TC) response to changes in dietary fat, and some men had TC levels that were nearly double the levels measured in other men [11]. Nevertheless, in 1953, Keys proposed the lipid–heart hypothesis in presentations and publications. The lipid–heart hypothesis assumed that high TC levels from consuming too much fat, saturated fat, and cholesterol would increase the deposition of cholesterol and fat into arterial walls, thereby increasing atherosclerosis and contributing to heart disease. Keys also assumed that reducing TC by lowering the intakes of total fat and cholesterol and replacing saturated fat with polyunsaturated fat would prevent coronary heart disease (CHD) [12,13].

### 2.1. ΔS: Conflation of Saturated Fat with Trans-Fat

Based on the results of his early feeding studies, Keys in 1957 proposed Equation (1), which related changes in TC (ΔChol.) with the intake of dietary saturated fat (ΔS) and polyunsaturated fat (ΔP) [11]:ΔChol. = 2.74 ΔS − 1.31 ΔP(1)

Although Keys was aware that the hydrogenation of fats increased saturation and produced *trans*-fats, he did not include *trans*-fats as a separate parameter but only considered the “degree of saturation of their constituent fatty acids” [14] using their iodine values. Thus, Keys ignored the physiological effects of *trans*-fats and just included them in ΔS. For some feeding experiments, Keys used commercial hydrogenated coconut oil (“Hydrol”) and reported an iodine value of 3. Since the iodine value of coconut oil is 6.3–10.6 [15], this indicates that Hydrol was partially hydrogenated and likely contained *trans*-fat [16]. It is significant that Keys noted a discrepancy in the equation for ΔChol. which he admitted “might be related to the trans acids in this hydrogenated fat (coconut oil)”, but he ignored their effects, as well as discrepancies that he would encounter later. It is worth noting that in 1957, Kummerow and co-workers had already developed a method for the quantitative measurement of *trans*-fat using infrared spectroscopy and reported the presence of *trans*-fatty acids in autopsy and biopsy material taken from human subjects. Kummerow also reported that hydrogenated shortenings and margarines contained 23 to 42% *trans*-fatty acids, as well as a complex mixture of many other geometric and positional isomers that formed during the hydrogenation process [17]. The extensive presence of *trans*-fat in the free diet at this time is exemplified by the 1957 paper of Malmros and Wigand who reported that serum cholesterol levels unexpectedly did not rise significantly in men when their “free” diet was replaced by a diet in which hydrogenated coconut oil was the sole fat providing 40 per cent of total calories [18]. However, this unexpected result may be explained by the presence of *trans*-fat (e.g., margarine) in the “free” diet as well.

In 1961, Keys and co-workers conducted a follow-up study in which subjects consumed 30 g of hydrogenated and unhydrogenated safflower oil or corn oil. They acknowledged that elaidic acid was present in the hydrogenated oils at up to 37% and that that “30 g of hydrogenated safflower oil in the daily diet produces a significantly higher serum cholesterol value than an equal amount of the natural oil”. They then stated that the result for ΔChol. agreed with what was predicted by the equation, with a minor adjustment [19]:ΔChol. = 2.68 ΔS − 1.23 ΔP(2)

While the equation worked for a diet with 30 g of hydrogenated safflower oil, it did not work well in other situations where ΔS had different amounts of *trans*-fat, as will be discussed below. Acceptance of the conflation of natural saturated fat and industrial *trans*-fat introduced a fundamental flaw in dietary research.

In 1961, the AHA in its first advisory on diet identified high intakes of total fat, cholesterol, and saturated fat as the primary causes of heart attacks [2]. With Keys as a co-author, the AHA advisory ignored the presence of *trans*-fats in the diet despite their full knowledge that “A considerable quantity of the fats and oils consumed in the United States are of the hydrogenated type”. This showed a clear disregard for *trans*-fat and the erroneous conflation of natural saturated fat and industrial *trans*-fat.

Between 1960 and 1980, deaths from CHD continued to increase steadily as the availability of animal fats (mainly butter and lard) dropped by half and consumption of polyunsaturated fat and *trans*-fat escalated [20]. Following the publication of the 1961 AHA advisory, the consumption of soybean oil in particular skyrocketed [21] in large part due to its use to produce hydrogenated shortening and margarine [22].

Since the labelling of *trans*-fat in the US was not mandated until 2006, most dietary surveys and epidemiological studies on saturated fat up to that year are likely tainted by the presence of *trans*-fat, unacknowledged and unaccounted for [23], unless specific measures were taken to exclude *trans*-fats from control and test diets, which was rare. The landmark Seven Countries Study, which was carried out in the US, Europe, and Japan, from 1957 to 1984, when *trans*-fats were widely available and unlabeled, concluded that “Death rates were related positively to average percentage of dietary energy from saturated fatty acids” [24,25]. However, a later analysis of the food consumed in the SCS revealed that the subjects consumed *trans*-fat: “Multivariate stepwise analysis selected butter, lard + margarine and meat as significant predictors and produced an R^2^ of 0.922”. Lard and margarine were combined “since in the 1960s most margarines were highly hydrogenated and resembling animal fat” [26]. Consistent with this observation, a study of dietary fats in Denmark reported that the average consumption during this period was more than 20 kg of margarine per person per year [27]. Thus, the mortality data that were attributed to saturated fat in the Seven Countries Study likely included the effects of *trans*-fat.

The *Dietary Guidelines for Americans* and other organizations, including the AHA and World Health Organization (WHO), list saturated fat and *trans*-fat together as fats to avoid as though they are equally harmful. However, there is an abundance of evidence that links the consumption of *trans*-fats with heart disease [28,29]. It is well known that the differences in the chemical structure between natural saturated fatty acids and man-made *trans*-fatty acids lead to profound biological and metabolic differences in how they perform. Hydrogenation is a process in which oils or fats are subjected to high heat and pressure using a catalyst along with the injection of hydrogen, which transforms unsaturated fat to saturated fat and *trans*-fat. The end-product can be solid, semi-solid, or liquid depending on the extent of hydrogenation and temperature. Partially hydrogenated oils (PHOs) may be liquid or semi-solid at 25 °C, whereas heavily hydrogenated fats melt at higher temperatures. The main purposes of hydrogenation are to prevent the oxidation of unsaturated bonds to prolong shelf life and to produce fats that convey a particular food texture and taste [30]. However, a consequential downside of hydrogenation is the formation of unnatural *trans* double bonds.

As an important example, consider the following C18 fatty acids: stearic acid (C18:0), oleic acid (C18:1^*9*-*cis*^), and elaidic acid (C18:1^*9*-*trans*^). Elaidic acid, which is more rigid than oleic or stearic acid, reduces the fluidity of cell membranes and affects the transport of substances in and out of the cell. The lipid composition of the cell membrane is markedly different in cells that are incubated with elaidic acid compared to those incubated with stearic or oleic acid [31].

*Trans*-fats are inflammatory, cause calcification of arterial cells, and can shorten the life of the cell [16]. Animal and epidemiological studies have linked the consumption of *trans*-fats to systemic inflammation, heart disease, cognitive disorders, Alzheimer’s disease, diabetes, obesity, non-alcoholic fatty liver disease, and cancer [28]. Some of the mechanisms of action of *trans*-fats which explain these effects have been elucidated [32,33].

Mary Enig was one of the earliest and most persistent voices to sound the alarm in the 1980s about the potential harms of replacing natural animal and vegetable saturated fat with industrial *trans*-fats. For her 1984 dissertation, Enig fed *trans*-fats to rats and reported that *trans*-fats interfered with enzyme systems that neutralized carcinogens, increased other enzymes that potentiated carcinogens, and caused obesity [34]. In later analyses of more than 220 items in 35 food types, Enig and co-workers found that previous publications had underestimated *trans*-fat consumption in the US and that many food labels underreported the amount of partially hydrogenated vegetable oil in the products. They found an average of 25.3% *trans*-fat in shortenings, 10.2% in salad and cooking oils, 23% in margarines, and up to 30% in potato chips, 37% in French fries, and 28% in fried chicken, and they estimated that the intake of *trans*-fat ranged from 1.6 to 38.7 g/person/day. This estimate was corroborated by measurements of *trans*-fatty acid isomers in human adipose tissue samples which ranged from 0.7 to 28.7 g per day. In 1990, Enig pushed for the mandatory labelling of *trans*-fat in foods sixteen years before this would become a reality in the US [35,36]. Enig co-authored a detailed historical review of the seed oil industry in the US which was published in two parts in *Nexus Magazine* in 1998 and 1999 [37].

The first edition of the *Dietary Guidelines for Americans* (*DGA*) in 1980 focused on lowering serum cholesterol to reduce heart disease by “avoiding too much total fat, saturated fat, and cholesterol”. The *DGA 1995* introduced limitations on total fat at 30% and saturated fat at 10% of total calories and mentioned the term “trans-fatty acids” for the first time, stating that “Partially hydrogenated vegetable oils, such as those used in many margarines and shortenings, contain a particular form of unsaturated fat (trans fatty acids) that is less effective than mono- or polyunsaturated fats in reducing blood cholesterol” [38]. This is a misleading statement: in fact, partially hydrogenated vegetable oils contain *trans*-fatty acids that *raise* serum cholesterol [39].

In 1995, the International Life Sciences Institute (ILSI), a group that was founded by an executive of the Coca Cola Company and largely financed by food and chemical corporations, commissioned an expert panel to study the health effects of *trans*-fats in the diet. The ILSI study concluded that *trans*-fat did not raise serum cholesterol levels as much as saturated fat, that the evidence of an increased risk of CHD from consuming *trans*-fat was inconclusive, and that more human trials were needed [40]. However, strong dissenting opinions were published in reaction to the report [41,42]. Also in 1995, the AHA launched a certification program that allowed food manufacturers to place the AHA Healthy-Heart check label on low-fat products and margarines with *trans*-fats [43]. The ILSI report strongly endorsed “fat-modified” food products: “The continued introduction of a wide variety of reduced-fat and fat-modified products into the marketplace should reduce both total fat and *trans* fatty acid intake. Since 1978 there has been more than a twofold increase in the number of adult Americans who consume such products. The Healthy People 2000 goal of having 5000 marketed products that are reduced in fat and saturated fat by the year 2000 has already been met: more than 5600 fat-modified products are now available”. Some of these low-fat products, which are high in refined grains and sugar, were developed in 1963 for the National Diet Heart Study (NDHS) which was funded by the NIH (see below).

However, the *DGA 2000* warned that “foods high in trans fatty acids tend to raise blood cholesterol. These foods include those high in partially hydrogenated vegetable oils, such as many hard margarines and shortenings. Foods with a high amount of these ingredients include some commercially fried foods and some bakery goods” [44]. The *DGA 2005* went farther with a substantial discussion of *trans*-fats, including a table of the *trans*-fat content of common foods and advice to “keep consumption as low as possible” but still limited saturated fat intake to less than 10% by making choices that are “lean, low-fat, or fat free” [45].

In 2003, the FDA announced that the labelling of *trans*-fat on food packaging would be required by 2006 [46], and in 2015, it released its final determination that PHOs were no longer generally recognized as safe (GRAS) for use in human food [47]. However, the FDA still permitted up to 0.5 g of *trans*-fat per serving size, a provision which still allows food products to contain industrial *trans*-fat.

From 1961, when the AHA published its first advisory, until 2006, when the FDA mandated the labeling of *trans*-fats, saturated fat and *trans*-fats were not separately labeled, making dietary studies on saturated fats inherently unreliable. The 1961 AHA advice to reduce total fat intake and avoid foods containing natural saturated fat, such as whole-fat dairy, eggs, and other animal fat, relied on animal studies and small metabolic ward studies and was premature, occurring before the first large-scale study in people had taken place. This guidance has not changed in the *DGA* despite a lack of evidence that heeding this advice will reduce a person’s risk of dying from heart disease. Furthermore, this guidance has promoted a persistent, unwarranted fear of consuming nutrient-dense foods which have been consumed by humans for many millennia, such as meat, eggs, dairy, and coconut. As a result, since the mid-twentieth century, the diets in the US have gradually shifted away from traditional whole foods toward imitation and ultra-processed foods which often contain high-fructose corn syrup, refined grains, synthetic vitamins, preservatives, and other additives, which are not nutritionally equivalent to whole foods. As a result, the US has experienced increasing rates of metabolic disorders in people of all ages, and it is reasonable to believe that this new epidemic is related to this major shift in dietary patterns. This will be discussed in greater detail below.

### 2.2. Iodine Value: Conflation of Plant-Derived Saturated Fat with Animal Fat

Another significant source of confusion regarding saturated fat is the conflation of animal fat with plant-derived saturated fat. Keys used the single parameter of iodine values to classify fats and oils, ignoring their cholesterol content. Iodine value is a chemical method for the estimation of the amount of unsaturation in a sample: the higher the iodine value, the higher the unsaturation in a fat or oil sample. Keys used iodine values to classify the fat and oil samples that he fed to his test subjects in his 1957 paper, and in 1965, he discovered that “the square-root of the iodine value is a reliable predictor of the serum cholesterol value” [48]. Keys then used the iodine values to define saturated fats as fats that raise serum cholesterol. This placed animal fat and plant-derived oils in the same category as saturated fat despite their very different SFA content. As shown in Appendix A, the amount of SFA in coconut oil is 82.3 g/100 g, while palm oil, lard, tallow, and butter have a much lower SFA content with 49.4, 38.9, 48.4, and 42.2 g/100 g, respectively. Further, plant-derived oils have no cholesterol, while animal fats contain high amounts of cholesterol. Keys ignored these compositions and used the single parameter of the iodine value to define saturated fat because the iodine value gave a linear relationship with serum cholesterol. The dietary guidelines adopted Keys’ definition of saturated fat as a fat that raises serum cholesterol. Early studies which used iodine values as criteria to classify fats and oils instead of fatty acid profiles should be reassessed. For example, lard is often used in dietary studies to represent all types of saturated fat despite the fact that the composition of plant-derived oils and animal fat are very different. The conclusions from studies that use lard as reference saturated fat should apply only to lard.

### 2.3. Solid Fat: Conflation of Plant-Derived Saturated Fat, Animal Fat, and Trans-Fat

A further source of confusion is the use of the term “solid fat” in food-frequency surveys and dietary guidelines. This term does not give a specific melting point temperature and it conflates plant-derived saturated fat and animal fat with solid margarines and semi-solid shortenings. Thus, “solid fat” is not a valid scientific description of saturation, nor does it indicate the content of *trans*-fat, but this term is used in the *Dietary Guidelines for Americans* [49].

### 2.4. ΔP: High-Linoleic-Acid Diet

Throughout the eight papers where Keys tried to develop a predictive equation, there was minimal discussion regarding polyunsaturated fat even though ΔP was present in all the equations. In his 1957 paper, Keys focused mainly on “polyethenoid” (linoleic acid) and ignored “monoethenoid” (oleic acid) because his experiments found very little effect of the latter on serum cholesterol. Keys completely ignored oleic acid and alpha-linolenic acid in most of his papers [11] and focused almost exclusively on linoleic acid [50]. It is ironic that one of the conclusions from Keys’ Seven Countries Study linked olive oil and oleic acid to low death rates from CHD but not linoleic acid [24].

The failure to consider the role of various polyunsaturated fats in the lipid–heart hypothesis may have led to the lack of guidelines regarding the consumption of omega-6 and omega-3 fat. Although linoleic acid and alpha-linolenic acid are essential fatty acids, the intake of PUFA is healthy only under three conditions. First, an excessive intake of linoleic acid has been linked to heart disease [51] and obesity [52]. Studies on rats [53] and humans [54] show that a linoleic acid intake of 4–7% of total energy is healthy, but excessive omega-6 consumption has been shown to be unhealthy because it is pro-inflammatory at high amounts. Omega-6 fatty acids promote vasoconstriction and blood clot formation, whereas omega-3 fatty acids generally have opposite effects [55]. Second, to avoid an imbalance in these effects, the ratio of omega-6 to omega-3 fatty acids should not exceed 5:1. The “Daily Nutritional Goals” tables for age groups and genders recommend a ratio of about 10:1 for omega-6 linoleic acid to omega-3 alpha-linolenic acid, much higher than the 1:1 to 5:1 ratios recommended based on age group [56]. Omega-6 and omega-3 metabolic pathways have enzymes in common, and excessive linoleic acid can interfere with the conversion of alpha-linolenic acid to docosahexaenoic acid (DHA) and eicosapentaenoic acid (EPA) [52]. Third, PUFA oils are unstable to heat and air and readily oxidize, producing degradation products such as *trans*-fatty acids, aldehydes, ketones, epoxides, hydroxy compounds, and free radicals which have been linked to heart disease [57,58,59]. Thus, dietary PUFA should be consumed in fresh food and should not be used for frying. However, many PUFA oils, such as soybean, corn, and canola oil, are advertised for use in frying. This introduces uncertainties in the results of dietary food-frequency surveys.

Dietary guidelines offer no recommendation regarding the amount of polyunsaturated fat that should be consumed in the diet and how to avoid the excessive intake of omega-6 fat.

### 2.5. ΔZ: The Unresolved Role of Dietary Cholesterol

Recommendations against dietary cholesterol were justified by studies conducted early in the 1900s in which extreme amounts of egg yolk or cholesterol relative to typical human intake were fed to rabbits, which are herbivores. This resulted in lesions that resembled atherosclerosis, as well as extensive damage to other organs [60]. The 1961 AHA advisory [2] cited these studies to support its recommendation.

In a four-part series of papers in 1965 [48,61,62], Keys revised his equation and added the new variable of dietary cholesterol to the calculation of serum cholesterol:ΔChol. = 1.2(2ΔS − ΔP) + 1.5ΔZ(3)

Here, ΔZ is the square root of dietary cholesterol in mg/1000 calories. This equation predicts that for every 1% increase in caloric intake of SFA, the serum cholesterol should rise by about 2.7 mg/dL. However, Keys himself noted that butter raised serum cholesterol levels by only 1.95 mg/dL at typical levels of consumption [61]. This equation has been used to formulate the *Dietary Guidelines for Americans*, even though various researchers have commented that the results of their studies did not agree with the predicted concentrations of serum cholesterol [63,64,65].

In 1984, Keys published his last paper [50] on his predictive equations where he tried to improve his regression formula to explain the disparate serum cholesterol data of test subjects from Minnesota and Massachusetts which came from two of the five “open” diet groups of the 1968 National Diet Heart Study (NDHS). Keys modified Equation (3) to obtain a solution to the data from the Minneapolis–St. Paul, Minnesota, test subjects:ΔChol. = 1.3(2ΔS − ΔP) + 1.5ΔZ(4)

However, the data from the Boston, Massachusetts, group required a different equation:ΔChol. = 2.16ΔS − 1.65ΔP + 6.77ΔC − 0.5(5)

Without explanation, Keys used a different parameter for dietary cholesterol in Equation (5) for the Massachusetts open cohort, where ΔC was measured as mg of dietary cholesterol per day. Mathematically, the units used for ΔZ and ΔC are not compatible. Keys admitted that the Minnesota equation underpredicted serum cholesterol by about 5% while the Massachusetts equation overpredicted it by about 300%. A possible reason for the discrepancy may have been due to the different amounts of *trans*-fat in the food products that were available in the two cities, which were masked by the use of ΔS. The accuracy of the equation for the three other open-center groups was not reported. Keys ended his paper with an admission that “This is not the place to speculate about the possible effect (of dietary cholesterol) on the risk of a heart attack or death from CHD”. In 1986, Hegsted re-evaluated the data on serum cholesterol responses to dietary cholesterol and concluded that “no predictive equation can explain such values” [66]. There were no further attempts to improve the Keys equation.

Improvements in clinical technologies have enabled the measurement of the lipoprotein cholesterol fractions of TC, in particular, very low density (VLDL-C), low density (LDL-C), small dense (sdLDL-C), intermediate density (iLDL-C), and high density (HDL-C) [67]. And just as Keys was unsuccessful in obtaining an equation to predict ΔChol., there is no general agreement on which specific fraction of lipoprotein cholesterol can accurately predict heart disease [68].

It is often claimed that saturated fats, in particular coconut oil, raise TC and LDL-C [69]. However, there are no reports that coconut oil actually causes heart disease although there have been no long-term clinical studies to determine this [70]. While a detailed discussion of the conflicting reports on the impact of coconut oil on TC and LDL-C is beyond the scope of this review, suffice it to say that there are clinical studies that report favorable effects of coconut oil on TC and LDL-C and that it raises HDL-C [71,72,73,74,75,76]. In a recent review of sixteen studies comparing the relative changes in the lipid profile of coconut oil versus other oils and fats, for groups consuming coconut oil, seven studies reported an average decrease in LDL-C, and most studies reported an increase in HDL-C [69].

Recommendations regarding the impact of dietary cholesterol on serum cholesterol have remained confusing and controversial. The answer may lie in the effect of other factors on serum cholesterol levels, such as total calorie intake, consumption of dietary fiber, carbohydrates, weight loss or gain ([77], Suppl. 3, pp. 1–428), lifestyle factors [78], and type of employment and level of education [79]. In 1950, a study which Keys himself co-authored recorded that serum cholesterol among normal males varied by age, from 174 mg/dL for 20-year-olds to 237 mg/dL for 65-year-olds [80]. This suggests that normal serum cholesterol levels also vary by age. Thus, the Keys equation that defines ΔChol. to be due only to ΔS, ΔP, and ΔC or ΔZ is erroneous.

There is a popular misconception that cholesterol and saturated fats are harmful substances, although both are critically important to life. Cholesterol is endogenously produced in all human cells as needed for numerous physiological processes. Cholesterol provides structural support within the membranes of cells and organelles as well as for the delicate neuronal networks of the brain and spinal cord. Cholesterol is a signaling molecule and a precursor for vitamin D, hormones, bile acids, and other substances [81]. Likewise, SFAs are metabolized to other lipids shortly after digestion and are also produced endogenously within cells from other fatty acids as needed to carry out many vital functions in the brain, lungs, and other organs [82]. Saturated fatty acids (SFAs) are generally divided into two groups according to their metabolism: medium-chain fatty acids (MCFAs, C6:0 to C12:0) and long-chain fatty acids (LCFAs, C14:0 to C18:0). Coconut oil is 54.5 g MCFA/100 g, while the amounts of MCFA in other fats and oils are very low: palm oil, 0.6; lard, 0.3; tallow, 0.9; and butter, 6.4 (see Appendix A). The physiological effects of the various SFAs also vary. MCFAs have been shown to support a healthy metabolism [83] and play an important role in the immune system [84]. However, the misconception that all SFAs have the same physiological effects continues until today, and many studies use C16:0 to represent all SFAs.

In response to the emerging scientific evidence on cholesterol, the warning in the *DGA 2010* to limit dietary cholesterol to below 300 mg per day was modified in the *DGA 2015,* but the warning remained: “The Key Recommendation from the 2010 Dietary Guidelines to limit consumption of dietary cholesterol to 300 mg per day is not included in the 2015 edition, but this change does not suggest that dietary cholesterol is no longer important to consider when building healthy eating patterns” [49]. In the *DGA 2020*, the warnings regarding total cholesterol, LDL-C, and dietary cholesterol were emphasized, but HDL-C was not mentioned even once ([3], p. 5). On the other hand, the 2020 AHA science advisory recommended healthy dietary patterns that are relatively low in cholesterol, such as a low-fat “Mediterranean-style” diet without mention of olive oil, encouraging instead the consumption of polyunsaturated liquid non-tropical vegetable oils [85].

Despite its numerous flaws, the lipid–heart hypothesis continues to permeate dietary guidelines. Although the presence of *trans*-fat in the food supply will now diminish, the errors that *trans*-fat caused as ΔS have not been corrected, and the warnings against natural saturated fat continue. High linoleic acid consumption, ΔP, continues to be promoted, and the warnings regarding total cholesterol, LDL-C, and dietary cholesterol are constantly repeated. These remain ingrained in the popular media and in the public mind. The lipid–heart hypothesis continues to be a central paradigm in dietary guidelines although there is abundant evidence that it is erroneous.

## 3. The Lipid–Heart Hypothesis Is Not Supported by Observational and Epidemiological Evidence

Numerous dietary studies were launched to prove the lipid–heart hypothesis by application of the Keys equation. Observational and epidemiological studies related to the lipid–heart hypothesis are discussed in this section, and the clinical studies are covered in the next section.

### 3.1. Framingham Multi-Generational Study

The Framingham Multi-Generational Study (1948–present) is a decades-long ongoing longitudinal observational study. A 1987 analysis of food records from 25% of the original cohort found no associations between TC levels and most aspects of dietary intake, including the percentage of calories as total fat or as animal fat, the ratio of plant to animal fat, or daily cholesterol intake. The average dietary intake of the people who did and did not develop CHD was the same. No aspect of diet correlated with the development of CHD, including daily intake of cholesterol, total fat, animal fat, PUFA/SFA ratio, or calories [86]. William Castelli, who was director of the Framingham Study from 1979 to 1995, lamented in a 1992 paper, more than 40 years after the study began, that “Most of what we know about the effects of diet factors, particularly the saturation of fat and cholesterol, on serum lipid parameters derives from metabolic ward-type studies. Alas, such findings within a cohort studied over time have been disappointing, indeed the findings have been contradictory. For example, in Framingham, Massachusetts, the more saturated fat one ate, the more cholesterol one ate, the *more calories one ate*, the lower the person’s serum cholesterol” [87].

### 3.2. Seven Countries Study (SCS)

The Seven Countries Study (SCS) (US, Europe, Japan, 1957–1984) was a 25-year longitudinal observational study that was designed and led by Ancel Keys [24]. There were 15 cohorts in the seven countries, namely, US, Italy, Greece, The Netherlands, Finland, Yugoslavia, and Japan. The SCS sought to look for associations between total and specific dietary fat and cholesterol intake, serum cholesterol levels, smoking, blood pressure, and death rates from CHD, cancer, and all causes in men consuming their usual diet. The SCS included 12,763 “healthy” men aged 40 to 59, although many had evidence of heart disease based on EKGs obtained at baseline. The participating countries and cohorts were not representative, and the dietary records were incomplete. For example, the US railroad cohort of men are not representative of the US population, and they only completed a single-day dietary record. Analysis of the diets did not distinguish animal fats from hydrogenated fats, which were widely consumed during this period [88]. The following additional results from the 15-year analysis were obtained [24]:There was no association of TC levels with CHD deaths.High PUFA intake had no association with coronary heart deaths: the cohort with the highest CHD death rate of 12/100 men consumed 2.9% PUFA, which was within the same range of 1.9 to 3.5% as the cohorts with the five lowest coronary heart death rates at ≤2/100 men.Although Keys ignored oleic acid in his previous studies, the results from the SCS showed that all-cause and CHD death rates were low in cohorts that consumed olive oil as the main fat.There was no association between the percentage of daily calories as total fat and all-cause deaths or CHD deaths (Figure 1A): Crete had the lowest all-cause and CHD deaths but one of the highest fat intakes at 36.1%. However, the East Finland cohort that consumed a comparable amount of fat at 38.5% had the highest coronary heart deaths. The six cohorts with the lowest CHD deaths had a total fat intake ranging from 9 to 36.1%.Keys claimed that there was an association between CHD deaths and the ratio of the intake of monounsaturated fatty acid to saturated fatty acid (MUFA/SFA). However, the data showed otherwise: two cohorts with the lowest CHD death rates (Tanushimaru and Ushibuka) had the same MUFA/SFA ratio of 1.0 as the cohort which had the second highest number of CHD deaths (US railroad men) (Figure 1B).A later analysis revealed that the food consumed in the Seven Countries Study included margarine (*trans*-fats) [26].

As mentioned above, the cohorts chosen represented very different types of employment and lifestyle, and US railroad men did not represent the US population. US railroaders would have been exposed to noxious oil and fuel emissions from locomotives much more than the general population, and 74% had reported ever smoking, compared to 42%, the all-time peak, in the general US population in 1960. Therefore, the results of the SCS for the US railroaders could not be extrapolated to represent the entire US population.

The 15-year report only provided data for all-cause, cancer, and CHD mortality and did not report information about the other causes of death. For example, regarding the two Japanese cohorts, the report pointed to low total fat (9%), low saturated fat (2.9%) consumption, very low TC levels, and low CHD death rates (144 and 127 per 100,000). However, by 15 years, the all-cause deaths were 1517 and 2013 per 100,000 men in the two Japanese cohorts compared to 1575 per 100,000 in the US men, who had the second highest CHD deaths (773 per 100,000). In addition, cancer death rates were much higher in the two Japanese cohorts (518 and 728 per 100,000) than in the US men (384 per 100,000). The other causes of death not reported were 57% of total deaths in the Japanese cohorts but only 26.7% in the US cohort. Cerebrovascular disease (stroke) death rates could have been reported but were not, and this was the leading cause of death in both Japan and Greece in 1960, responsible for about 30% of total deaths in both countries but caused many fewer deaths in the US at that time, about 108 per 100,000, and was the third leading cause of death behind heart disease and cancer [89]. Japan had the lowest baseline cholesterol levels among all cohorts in the SCS with decile averages (10th to 90th percentiles) by age group ranging from 109 to 277 mg/dL in the cohort from Tanushimaru and from just 99 to 204 mg/dL in the cohort from Ushibuka. This compared to values ranging from 182 to 298 mg/dL in the US cohort [88]. In a book about his adventures as an investigator in the SCS, Henry Blackburn, who was a long-time Keys collaborator, stated that “Ancel Keys and colleagues hypothesized that the Japanese, at the lower pole of fat diets, would hinge the correlation between saturated-fatty-acid intake and coronary disease risk”. Blackburn noted that the Japanese diet was low in animal fats, except fish, and high in complex carbs with several times the salt consumption of most other traditional diets; the rates of cerebral hemorrhage were quite high. Blackburn also reported that as economic conditions and the diet improved in Japan, cerebral hemorrhages subsided substantially, which Japanese investigators in the 1990s attributed as much to the increase in cholesterol levels as to lower average blood pressure levels, since rates of cerebral hemorrhage also decreased in other cultures whose blood cholesterol values had increased but did not consume so much salt [90].

The SCS investigators were intent on proving the lipid–heart hypothesis and the value of the Keys equation. Although an extraordinary amount of data was collected on health risks and other causes of death, the major focus was on CHD. Blood vessels in the heart and brain are subjected to similar stresses and pathological processes, and it is surprising that they reported associations of many suspected risk factors with CHD and not with stroke, which was a much more prominent cause of death than CHD in some of the SCS countries, in particular, Japan and Greece. In addition, since data were collected for each subject, it would have been helpful to determine whether there were associations of any dietary factors and other risk factors for the men who died from heart disease, cancer, and other causes compared to those who did not die during the study period, but no such information was provided in the reports. In summary, the SCS failed to prove that the consumption of natural saturated fat is linked to heart disease.

### 3.3. A Study from the National Cholesterol Education Program (NCEP)

In the latter part of the 20th century, the focus of attention narrowed to the largest lipoprotein fraction, identifying elevated LDL-C levels as a likely culprit in coronary artery disease (CAD). However, a study from the National Cholesterol Education Program (NCEP), which was developed by the National Heart, Lung, and Blood Institute of the NIH, suggested that an association between LDL-C and CAD might not be so strong. This study looked at lipid values at the time of hospitalization for 136,905 people admitted to 541 hospitals with confirmed diagnoses of CAD, including acute coronary syndromes, CAD requiring a revascularization procedure, or other CAD diagnoses unrelated to heart failure. The mean admission LDL-C level was 104.9 mg/dL with 75% below 130 mg/dL and 17.5% below 70 mg/dL, while 54.6% had abnormally low admission HDL-C levels of <40 mg/dL. However, 21.1% of patients were taking lipid-lowering agents before admission [91].

Numerous other studies have shown that LDL-C is not a reliable predictor of heart disease and that inflammation, the oxidation of cholesterol, and small dense LDL-C particles (sdLDL-C) may be more important factors [92,93]. These findings support the calls to revise the dietary guidelines which recommend lowering TC and LDL-C levels by replacing saturated fat with polyunsaturated fat, while ignoring the benefit of HDL-C.

### 3.4. Observational and Historical Evidence on Coconut Oil, a Saturated Fat

Coconut oil is made up of over 85 g saturated fat per 100 g of oil and contains no cholesterol [94]. Compared to all other common dietary fats and oils, coconut oil contains the highest amount of saturated fat on a weight basis. Coconut oil makes up about 35% of the fresh weight of the kernel and 27% in coconut milk [95]. Observational and historical reports on the health effects of coconut may vary depending on the specific coconut intake [96]. The Pukapuka and Tokelau island studies considered the whole coconut diet. Pukapuka and Tokelau are among the most isolated islands in the Pacific. Foreseeing that the Western diet would alter the traditional diet of the Pacific islanders, Ian Prior undertook a study on the health status of the inhabitants who consumed large amounts of saturated fat in their coconut-based diet. Prior published two papers, the first in 1973 [97] and a follow-up in 1981 [98], where he reported that “vascular disease is uncommon in both populations and that there is no evidence of the high saturated fat intake having a harmful effect in these populations”. Prior’s study was prescient because, upon the arrival and adoption of the Western diet, the people in the Pacific islands became afflicted with obesity, diabetes, heart disease, and other ailments that are common in Western countries. In 2003, the World Health Organization (WHO) Western Pacific Region reported that people from most islands in the Pacific were “2.2 times more likely to be obese and 2.4 times more likely to be diabetic if they ate imported fats than if they ate traditional fat sources” [99]. Similar observations were made on the native Hawaiians who were healthy and fit consuming their traditional diet before the entry of the Western diet [100]. A number of observational studies from coconut-consuming countries, such as the Philippines [101], India [102], and Indonesia [103], have reported no link between the coconut diet and heart disease. A two-year randomized study in India on 200 patients with stable CAD comparing coconut oil and sunflower oil as cooking media reported that there were no statistically significant differences in the anthropometric, biochemical, vascular function, and cardiovascular events between the two groups after 2 years [104]. A meta-analysis of observational evidence on the health effects of coconut oil concluded that the consumption of coconut oil in traditional diets does not lead to adverse cardiovascular outcomes [70].

### 3.5. Prospective Urban Rural Epidemiology (PURE) Study

One of the largest, and certainly the most representative, long-term epidemiological studies ever conducted is the Prospective Urban Rural Epidemiology (PURE) study. The PURE study was conducted in eighteen countries on five continents worldwide, which included three high-income countries (Canada, Sweden, and United Arab Emirates), eleven middle-income countries (Argentina, Brazil, Chile, China, Colombia, Iran, Malaysia, occupied Palestinian territory, Poland, South Africa, and Turkey), and four low-income countries (Bangladesh, India, Pakistan, and Zimbabwe). After analyzing data from 135,335 participants aged 35 to 70 at enrolment, who were followed for an average of 7.4 years, 5796 deaths and 4784 major cardiovascular disease events were reported. People in the highest quintile of carbohydrate intake had 1.28 times the risk of dying prematurely compared to those with the lowest intake, although there was no significant association with cardiovascular disease events or related mortality. However, the highest quintiles of fat intake compared to the lowest intakes were associated with a lower risk of premature death from all causes (total fat hazard ratio (HR) 0.77, saturated fat HR 0.86, MUFA HR 0.81, and PUFA HR 0.80) and were not significantly associated with major cardiovascular disease (total fat HR 0.95, SFA HR 0.95, MUFA HR 0.95, PUFA HR 1.01) or related mortality (total fat HR 0.92, SFA HR 0.83, MUFA HR 0.85, PUFA HR 0.94). People at the highest quintile of saturated fat intake had a significantly lower risk of stroke compared with the lowest quintile (HR 0.79) [105]. A more detailed study on the consumption of dairy revealed that a higher intake of total dairy (>2 servings per day compared with no intake) was associated with a lower risk of total mortality, including cardiovascular mortality [106]. The result of this large global dietary study calls for reconsideration of the dietary warning against saturated fat as well as recommendations to avoid whole-fat milk, which first appeared in the 1961 AHA advisory on dietary fat and heart disease and have continued as guidance in every version of the *DGA* since 1985.

## 4. Other Factors That Contributed to the Increase in Coronary Heart Disease

Although dietary fat became the major research focus to explain the perceived increase in CHD beginning in the early 1900s, other important factors that determined health outcomes were overlooked. These include public health measures and various behavioral and environmental risk factors.

### 4.1. Public Health Measures

By 1900, US public health measures, including hygienic handwashing as a medical practice, milk pasteurization, chlorination of water, and vaccination, had so significantly reduced infection-related mortality from the top three causes (pneumonia/flu, tuberculosis, and diarrheal illnesses) that heart disease moved from the fourth to the leading cause of death between 1900 and 1910, thus attracting attention. The US population exploded from 76.3 million to 158.8 million between 1900 and 1950, and the all-cause death rate in the US plummeted by 41.3% from 1641.5 per 100,000 people in 1900 to 963.8 per 100,000 by 1950 [89]. This contributed to a steady increase in life expectancy at birth in the US (except during the 1918–1920 Spanish flu epidemic), from 47.3 years in 1900 to 68.3 years by 1950 [107]. Between 1900 and 1950, total death rates from all causes declined in all age groups: by 81% in newborns to 34-year-olds, by 40.7% in 35- to 64-year-olds, and by 23.7% in people 65 and older. Also, from 1900 to 1940, deaths attributed to “diseases of the heart” declined by 71.8% in newborns to 34-year-olds combined but increased by 13.5% in people 35–44 years old, by 63.1% for people reaching ages 35 to 64, and by 187.8% for people 65 years and older (see Figure 2).

Up until the 1940s, vital statistics were published for just a few subcategories of “diseases of the heart”, such as “organic diseases of the heart” and “angina pectoris”, but in 1949, many more separate subcategories were added for “arteriosclerotic heart disease, including coronary disease”, “chronic rheumatic heart disease”, and “hypertensive heart disease”, for example [108,109]. So, recognition of arteriosclerotic heart disease and coronary disease as specific causes of death was relatively new when the lipid–heart hypothesis was introduced by Keys in 1953. Thus, due to marked reductions in deaths from infections and heart diseases in younger people, millions more were now surviving infancy through young adulthood and living into middle and old age, finally succumbing to heart disease and other causes much later in life.

While most people hope to avoid dying prematurely, it is inevitable that all will die, which then poses an existential question: if not from heart disease, then what would be a better alternative? In 1950, while diseases of the heart ranked highest, the other causes of death ranked from highest to lowest were cancer, vascular lesions affecting the central nervous system, accidents from all causes, certain diseases of early infancy, pneumonia/influenza, tuberculosis, general arteriosclerosis, nephritis/other renal sclerosis, and diabetes mellitus. Diarrheal conditions were no longer in the top ten causes of death. Though smoking, air pollution, and possibly *trans*-fats were also likely contributors, the perceived epidemic of coronary artery and other heart diseases in middle-aged men in the 1950s more likely reflected the much longer life expectancy and reductions mainly in infection-related causes of death than consumption of natural animal and vegetable saturated fats. The consumption of natural saturated fat was declining steadily as the consumption of industrial *trans*-fats was increasing in parallel to the increase in deaths from heart disease. This suggests that, if fats were a factor, *trans*-fats were more likely to be responsible (see Figure 3). Deaths from all causes had dropped dramatically, and the seeming epidemic of heart disease in middle-aged men might have been more perception than reality.

### 4.2. Behavioral and Environmental Risk Factors: Air Pollution, Smoking, Hypertension, and Diabetes

Outdoor and indoor air pollution related to industrialization, urbanization, vehicle emissions, and tobacco smoking, which are known risk factors for heart disease, worsened throughout the 20th century. Between 1910 and 1940, cancer death rates had increased by 88.0% for all ages combined, and within each age group, more than doubling for people aged 65 and older. Lung cancer was considered a rare malignancy in 1920 but began to skyrocket around 1930, becoming the leading cause of cancer deaths for men around 1953 and for women in the late 1970s [110,111]. Tobacco smoking in US adults increased steadily from 1905 until levelling off in the 1960s, then dropped from a peak of 42.4% to 24.7% in 1997 following the release of the Surgeon General’s 1964, “Report on Health and Smoking” [112]. With the decline in tobacco use, coronary artery deaths also trended downward in the 1980s. A CDC report on public health advances in the 20th century cited the fall in tobacco use but did not mention changing dietary fat intake as a contributing factor for this improvement in heart disease deaths [113]. When deaths from heart disease were peaking in the 1960s, cigarette smoking was at an all-time high in the US (see Figure 4), and the consumption of margarine and shortening, which contained *trans*-fat, exceeded that of butter and lard (see Figure 3 above).

In addition, the Framingham Multi-Generational Study reported at its ten-year point in 1957 that high blood pressure and diabetes were major risk factors for CHD and that HDL-C had an inverse relationship with CHD [114], and no association of CHD with dietary factors was found in a later analysis [86]. Thus, the increase and decrease in heart disease mortality could be explained by other factors, in particular, longer lifespan, air pollution, smoking, hypertension, and diabetes, rather than the lipid–heart hypothesis.

## 5. The Lipid–Heart Hypothesis Is Not Supported by Clinical Studies

This section will discuss large long-term interventional clinical trials and autopsy studies of dietary fats and the lipid–heart hypothesis which were conducted between 1963 and 1980.

### 5.1. The Anti-Coronary Club Study

In the mid-1950s, Irvine Page, Norman Jolliffe, Ancel Keys, Fredrick Stare, Jeremiah Stamler, and others devised a diet, which was called the “Prudent Diet” [115], to test the lipid–heart hypothesis. This diet, which was introduced to the American public in 1956 in a nationally televised fundraiser for the AHA, involved a reduction in total fat and the replacement of lard, butter, cream, whole-fat milk, meat, and eggs with corn oil and other seed oils, margarine, skim milk, chicken, and cold cereal. The diet also encouraged the consumption of more fruit, vegetables, and nuts.

The Anti-Coronary Club Study (1957–1966) was a 10-year study of the Prudent Diet in 1,242 men aged 40–59 years with and without diagnosed heart disease. The test group was instructed to avoid the consumption of hydrogenated fats, while the control group consumed their usual diet which likely included *trans*-fats. The study reported mixed results with fewer heart attacks but more deaths in the Prudent Diet group than the control group, which experienced no deaths [116,117].

### 5.2. National Diet Heart Study

The National Diet Heart Study (NDHS) (US, 1963–1965) was conceived as a pilot study by the Executive Committee of the AHA “to test the hypothesis that alteration of amount and type of fat and amount of cholesterol in the diet would decrease the incidence of first attacks of clinical coronary heart disease in middle-aged American men” ([77], Suppl. I, p. I-1). Cardiologist Irvine Page, president of the AHA, received a grant from the National Heart Institute to conduct a study of the lipid–heart hypothesis using “fabricated fat-modified foods”, which would allow for the blinding of the participants as to what diet they were consuming. New foods were created to have similar taste, smell, and texture but differ in ratios of SFA, PUFA, and margarine. The study was designed and carried out by Keys, Page, and other investigators, and additional support came from AHA fundraising, other organizations, and private companies ([77], Suppl. 3, pp. 1–428).

The first study, which lasted 12 months, included middle-aged free-living men in five open centers and men living in a closed center at the Faribault State Hospital, a public residential facility serving people with intellectual disabilities. Two test diets (B and C) were low in saturated fat (≤9%) and high in PUFA (≥15), with 350–450 mg daily of cholesterol; Diet B and Diet C had 30% and 40% of total calories as fat, respectively. The control Diet D was designed to represent the typical American diet with 40% fat, ≥18% saturated fat, ≤7% PUFA, and ≥650 mg cholesterol/day. The Diet E group at Faribault Hospital consumed a very high PUFA/SFA ratio of 4.4. Diets B, C, and F were low in SFA (<9%) and high in PUFA (>14%), which presaged the *DGA* recommendations. The Diet X group received dietary instructions only and no provided foods (Appendix A). All men were instructed to remove or greatly reduce foods with natural saturated fat, including egg yolks, full-fat milk, butter, and cheese, and to consume only skim milk, lean meats, and to trim the fat off meat, to ensure that most of the fat would be in the provided foods. For the other test diets, at least thirty food manufacturers were recruited to create special foods which included oil-filled sausages and patties, imitation eggs, imitation ice cream, imitation cheese loaves, coffee creamers with hydrogenated oils, margarines, cakes, pastries, and oil emulsions to replace natural food items, such as dairy fat. The food preparers at the special centers were provided with a table of oils and fats to create the special foods using ratios of SFA, MUFA, and PUFA that were adjusted according to the diet the men were assigned to, including heavily hydrogenated fat to represent saturated fat and PHOs to represent PUFA fats. Coconut oil was not used because its fat content could not be modified or controlled. All groups received *trans*-fats, and sources of natural saturated fat were intentionally removed from all diets. Fatty acid compositions were determined by gas chromatography, but the analysis did not include *trans*-fatty acids even though these were part of the NDHS diet ([77], Suppl. 1, pp. I64–I71).

The specific average TC results for each diet are provided in Appendix A. During the first 12 months, average TC levels decreased in the control group but even more so in the test diet groups with maximum reductions at 2 and 6 weeks but trended upward thereafter. The results for Diets B (30% total fat) and C (40% total fat), which differed by less than 1% at nearly all time points, contradicted the hypothesis that reducing total fat intake should have the effect of reducing TC levels. The men in the Diet X instructions-only group had the largest reduction in TC. For the open centers combined, the mean standard deviations *within* individuals for TC were about ±12.0 mg/dL and the mean standard deviations *between* individuals for TC ranged from ±33.0 to ±42.3 mg/dL, illustrating the marked variability in TC response within and between individuals. This has been reported in many other dietary fat and cholesterol studies, including Keys’ early studies. As a result, Keys and others concluded that such studies cannot predict the response of a given individual.

For the First Study, the Faribault Hospital closed center had comparable results to the open centers in initial TC results, and the levels were largely maintained, thereby illustrating the differences in response to an intervention between free-living people and confined individuals consuming the same strictly controlled diet. The men consuming Diet E, which had a very high PUFA/SFA ratio, had the largest reduction in TC, thereby demonstrating that replacing saturated fat with PUFA to this degree can have a strong cholesterol-lowering effect.

The diets were reformulated and renamed for the Extended and Second Studies at the open centers to study the effects of different ratios of PUFA/SFA ranging from 0.4 to 3.0 in diets containing about the same amounts of dietary cholesterol and total fat. The diet designs and average TC results of the Extended and Second Studies are provided in Appendix A. At the open centers, the TC results were similar for men consuming the diets with ratios of PUFA/SFA between 1.5 and 3.0; however, there was little difference in average TC results between the men consuming Diet G with 10% saturated fat and Diet D with 18% saturated fat. As expected, TC levels did not decrease in the true control Diet Z group.

It would have been helpful to determine the relative effects of natural saturated fat and *trans*-fat in TC results for the men consuming each of the diets in the NDHS, but these fats were not considered separately in the diet designs or reported separately in the diet chemical analyses.

The goal of the NDHS was to test Keys’ lipid–heart hypothesis and the Keys equation by demonstrating the effects on TC levels of different percentages of total fat and ratios of PUFA to SFA. However, the diets were designed for weight loss with a total energy intake of 400–600 kcal less than the men’s pre-study diets, which introduced an additional confounding variable: most subjects initially lost weight, which was already known at that time to reduce TC levels. The men with the largest weight loss had the largest reductions in TC. However, many regained the weight by the end of the study, and TC levels also rose in most cohorts. The NDHS report noted that, when people in previous metabolic ward studies gained weight, the average TC rose sharply, and TC decreased with weight loss, but once the weight stabilized at the new weight, the TC level tended to return to the baseline levels [118,119,120,121]. Regression analyses of the First Study results found that “There was an apparent effect of both recent and remote weight change on TC levels—even after [the] effects of reported dietary fats had been allowed for” ([77], Suppl. 1, p. I-212). In other words, the reductions in TC could have been largely due to weight loss.

Altogether, 1807 men, who were followed for six to eighteen months, completed the study. The investigators stated that the NDHS pilot study was not powered to look at CHD outcomes; however, they reported that, by the end of the three studies, eleven men experienced cardiovascular events, five from the control Diet D and six from the other test diets. Only one man, whose diet group was not reported, died. The results from the three stages of the NDHS suggest that the typical American diet and the created diets, all of which contained significant amounts of *trans*-fats, produced comparable cardiovascular outcomes. Two different equations, which Keys would later use in his 1984 paper, were developed to predict serum cholesterol, ΔChol., from natural saturated fat, ΔS (which included solid margarine and shortening), polyunsaturated fat, ΔP, and dietary cholesterol, ΔC or ΔZ (see Equations (4) and (5) above). However, the predicted ΔChol. values from two Keys equations differed significantly from the observed TC results.

Six months after the study was completed, 253 men at one open center had TC levels analyzed and had essentially returned to their original baseline values in all diet groups, which suggests that the effect of manipulating dietary fats may be temporary and that an individual’s metabolism readjusts over time to one’s genetically determined set point. Many dietary fat studies have a duration of only 2 to 6 weeks based on the assumption that TC levels stabilize by that time. However, the tendency for TC in all NDHS diet groups to trend upward by 12 weeks contradicts that assumption.

The expensive NDHS design based on Keys’ lipid–heart hypothesis greatly reduced the intake of natural saturated fat in all diet groups and replaced much of this with industrial *trans*-fats. The NDHS also showed that a low-fat diet which contains *trans*-fats does not reduce serum cholesterol levels or cardiac risk more than a high-fat diet, and it did not prove that replacing saturated fat with polyunsaturated fat reduces deaths from CHD. The study did not receive funding to move forward with the 100,000-man study. The NDHS did, however, fund the development of many new fabricated low-fat processed foods, some of which are still on the market today.

### 5.3. Multiple Risk Factor Intervention (MRFIT)

The Multiple Risk Factor Intervention (MRFIT) study (1971–1980) was a 10-year study of 12,866 high-risk men who were smokers with high blood pressure and high TC levels but without clinical evidence of heart disease at baseline. The control group continued their usual diet and local care, while the test diet group received intensive instruction on a low-fat (<35%), low-saturated-fat (<8%) diet with 2 to 4 tablespoons per day of margarine and high-PUFA oils. TC and LDL-C dropped more in the test diet group, but there were no significant differences in CHD events or deaths. However, the percentage of deaths from cancer was significantly higher for the test diet group (30.6%) than the control group (26.5%). The investigators expressed concern that the large amount of PUFA oils might have been toxic to the men [122,123].

### 5.4. Studies Cited in the 2017 AHA Advisory on Dietary Fats and Cardiovascular Disease

In 2017, the AHA published a presidential advisory on dietary fat and heart disease [124] which still promoted the tenets of the 1956 Prudent Diet. The advisory reviewed four “Core Trials on Replacing Saturated with Polyunsaturated Fat” which were conducted between 1968 and 1979: the British Medical Research Council Study [125]; the Dayton study [126]; the Oslo Diet-Heart Study [127]; and the Finnish Mental Hospital Study [128]. However, the results should be questioned due to the likely presence of *trans*-fats in some of the studies. In particular, the control group in the Finnish Mental Hospital Study were provided margarine. The Oslo Diet-Heart Study did not describe the diet of the control group, while the British Medical Research Council Study used animal fat exclusively. In the Finnish Mental Hospital Study, many patients received drugs that are now known to cause cardiac abnormalities on EKG, arrhythmias, and sudden death, the same criteria used to determine effects of the changes in dietary fat.

The Finnish study found no association of dietary fat with adverse cardiac outcomes and stated that there were too many variables to be able to ascribe any changes to a single factor. All four studies reported a reduction in serum cholesterol levels when saturated fat was replaced with polyunsaturated fat, but all four studies also reported that there were no statistically significant differences in total mortality, CHD events, or CHD deaths, even after years on the test diets. In addition, all four studies concluded that there were no apparent effects of a change in dietary fat on these outcomes.

A proper evaluation of the impact of saturated fat on serum cholesterol cannot be completed from the data reported by the four core studies. Nevertheless, using Keys’ 1984 equations which gave conflicting results, the AHA reported that there was a reduction in serum cholesterol levels when saturated fat was replaced with polyunsaturated fat and estimated that dietary cholesterol accounted for 15–20% of the reduction in serum cholesterol. The AHA assumed that the amount of *trans*-fat consumed was inconsequential and claimed that their own meta-analysis showed a significant reduction in cardiovascular disease, despite the conclusions of the authors of the four studies that there was no such reduction in CVD. The AHA advisory also cited the 2015 Cochrane review [129] which analyzed 15 randomized controlled trials conducted from 1965 to 2006. However, unlike the AHA, the Cochrane review admitted that it “could not explore data on trans fats”.

### 5.5. Sydney Diet Heart Study (SDHS)

The Sydney Diet Heart Study (SDHS) conducted in Sydney, Australia (1967–1973), involved 458 men aged 30 to 49. The men in the control group (n = 237) maintained their usual diet including the margarines that they were already consuming. Those in the safflower oil group (n = 221) were instructed to use only oils, margarines, and shortenings made from safflower oil in place of all foods containing “saturated fat”, including animal fat, butter, other margarines, salad dressings, baked goods, and shortenings. They were also instructed to increase PUFA intake to about 15%, to reduce saturated fat to less than 10% of total calories, and to limit dietary cholesterol intake to less than 300 mg/day. Safflower oil is about 90% omega-6 linoleic acid. Despite a larger decrease in TC in the high-PUFA safflower group (−37.4 mg/dL) compared to control (−15.5 mg/dL), the men consuming the safflower oil diet had significantly higher rates of all-cause death (17.6% vs. 11.8%, HR 1.62), cardiovascular deaths (17.2% vs. 11.8%, HR 1.70), and fatal CHD (16.2% vs. 10.1%, HR 1.74). However, the results of the SDHS were not reported by the original researchers but were published in 2013, 40 years later, from recovered raw data [130]. An updated meta-analysis in the same article showed no evidence of cardiovascular benefit from the replacement of saturated fats with polyunsaturated fats.

### 5.6. Minnesota Coronary Experiment (MCE)

The Minnesota Coronary Experiment (MCE) (US, 1968 to 1973), which was led by co-principal investigators Ancel Keys and Ivan Frantz, was expected to demonstrate the benefit of a high-PUFA diet. The MCE included 9,423 men and women with ages ranging from 20 to 97 years, living in nursing homes or mental hospitals, divided into a nearly equal number between the corn oil group and control group, who consumed the regular hospital diet with 18.5% saturated fat but also included hydrogenated and partially hydrogenated fats and oils. Corn oil replaced saturated fat as the cooking oil, and margarine replaced butter. Corn oil contains about 54% linoleic acid which increased omega-6 intake of the subjects by almost three-fold to 13.2% of total calories and reduced saturated fat by half to 9.2%. As anticipated, the average serum cholesterol level decreased from baseline much more in the corn oil group (−31.2 mg/dL) than in the control group (−5.0 mg/dL), but there were more cardiac events and deaths in the corn oil group. People with the largest reduction in TC had the highest incidence of death.

Unfortunately, the results of the MCE study were not published after its completion in 1973. Instead, partial results were published in 1989 by Frantz as principal author without Keys as co-author [131]. Frantz reported that there were 27.2 incidents of MI and sudden death per 1000 person-years for the corn oil and 25.7 incidents for the control group, and total deaths were 55.8 per 1000-person years for the corn oil group and 52.6 for the control group.

A more complete analysis of the MCE study was conducted by Ramsden and co-workers in 2016 from recovered raw data [132]. They calculated that for the 2,355 subjects who consumed the corn oil diet for more than one year, there was a 22% higher risk of death for each 30 mg/dL reduction in serum cholesterol and no reduction in coronary atherosclerosis or myocardial infarcts compared to the control group. Detailed autopsy reports on 149 subjects showed that 41% of those in the corn oil group had evidence of at least one myocardial infarction compared with only 22% in the control group. In addition, the corn oil group did not have less atherosclerosis in the coronary arteries and aorta than the control group, and “there was no association between serum cholesterol and myocardial infarcts, coronary atherosclerosis, or aortic atherosclerosis in covariate adjusted models”. An accompanying systematic review of five RCTs of replacing saturated fat with high-linoleic vegetable oils in 10,808 participants also found no benefit on mortality from CHD or all-cause mortality and no benefit for the prevention of non-fatal myocardial infarctions or for fatal and non-fatal myocardial infarctions combined. Ramsden and co-workers concluded that “Available evidence from randomized controlled trials shows that replacement of saturated fat in the diet with linoleic acid effectively lowers serum cholesterol but does not support the hypothesis that this translates to a lower risk of death from CHD or all causes”.

The Sydney Diet Heart Study and the Minnesota Coronary Experiment highlight two important points: First, serum cholesterol levels are not reliable biomarkers for heart disease—and may, in fact, predict the opposite. And second, both studies show that the replacement of saturated fat with high amounts of linoleic acid increases the incidence of heart disease. These two studies disproved Keys’ lipid–heart hypothesis. Had the results of both studies been published before 1980, the authors of first edition of the *Dietary Guidelines for Americans* would have been better informed.

### 5.7. Some Autopsy Studies Do Not Support the Lipid–Heart Hypothesis

The lipid–heart hypothesis has been refuted by many autopsy studies in people of all ages. The fetus and newborn already have areas of atherosclerosis despite very low serum cholesterol levels. Atherosclerosis may be a mechanism that protects and fortifies areas of arteries, such as points of branching, which are subjected to high pressure. Atheromatous plaques appear to result from inflammation, infection, and other damage to arterial walls [93,133]. Further supporting the autopsy findings reported in the MCE [132], Dayton and co-workers [126] reported no differences in autopsy studies in the degree of atherosclerosis or numbers of atheromatous plaques in the men consuming a high-PUFA diet versus a control diet. They also found that there were no differences between the high-PUFA diet and control groups in the percentages of triglycerides, free fatty acids, cholesterol, cholesterol esters, or phosphatides in the aorta, coronary arteries, or atheromatous plaques or in the amount of calcium in the aorta. In summary, human autopsy studies showed that the degree of atherosclerosis and atheroma formation are the same in the following scenarios:In people who had evidence of myocardial infarctions and those who do not [132];In people with high versus low serum cholesterol levels [133];In people who eat higher versus lower percentages of energy as polyunsaturated fat [126,132];In people who replace saturated fat with polyunsaturated fat [126,132].

Thus, autopsy findings have failed to confirm that replacing saturated fat with polyunsaturated fat reduces the degree of atherosclerosis or atheroma formation.

## 6. The Perceived Epidemic of Heart Disease Has Been Replaced by a Much Larger Epidemic of Metabolic Disorders

The lipid–heart hypothesis and the dietary fat studies that followed appeared in response to a perceived epidemic of coronary artery deaths, mainly in middle-aged men. Since the promulgation of the low-fat, low-saturated fat recommendation in 1961 by the AHA, the heart disease epidemic has been replaced by a much larger epidemic of metabolic disorders affecting all ages and genders, accompanied by staggering increases in the rates of obesity, diabetes, autism, dementia, and many other chronic diseases. In US adults over 20 years old, the percentage who were overweight or obese increased from 48.8% (0.9% severely obese) in 1960–1962 to 82.3% (9.2% severely obese) in 2017–2018. There was a particularly striking increase in adult obesity between 1980 and 2000 during the time that the first four editions of the *DGA* were published along with the first Food Pyramid in 1992 which encouraged six to eleven servings of grains daily and minimal fat intake [134]. In 1971–1974, 16.4% of US children and adolescents aged 2 to 19 were overweight (10.2%), obese (5.2%), or severely obese (1%), but by 2017–2018, this figure had more than doubled to 41.5% who were overweight (16.1%), obese (19.3%), or severely obese (6.1%) (see Figure 5) [135].

It is reasonable to assume that human milk is designed to promote the optimal growth and development of the brain and other organs in the rapidly growing infant and young child. A study of 64 traditional cultures conducted in the 1950s reported breastfeeding in some populations up to age 4 or 5, with an average age of weaning estimated at 2.8 years [136]. However, the CDC reported that, in 2019, by six months of age, just 55.8% of infants received any breastmilk and only 24.9% were exclusively breastfed, and therefore, most US infants receive substantial infant formula during the first 1 to 2 years of life [137]. USDA FoodData Central reports that human milk contains about 56% of total calories as fat, including 26% of total calories as saturated fat, 15% as MUFA, and 4.5% as PUFA. By comparison, the three leading infant formulas in the US contain on average 47.5% of calories as fat, with 18.7% as saturated fat, but more PUFA (9.8%) than human milk. Infant formulas contain considerably more linoleic acid (8.6%) than human milk (4.8%). It is noteworthy that most commercial infant formulas contain coconut oil to provide the medium-chain fatty acids and myristic acid which are found in human milk but not found in most other common vegetable oils [138].

The *DGA 2020* advised an abrupt reduction in total fat to 30–40% and saturated fat intake to <10% of total calories when a child turns two years old at a time when the brain and body are rapidly growing and developing. Evidence to support this guidance in long-term clinical trials is lacking. The USDA’s My Plate for Preschoolers, which is based on the *DGA 2020*, shows no fat on the plate and advises that even very young children eat only fat-free or low-fat dairy and lean meats [139]. This gives parents the impression that fat is unhealthy for children. With the protein requirement remaining constant, to provide adequate calories for growth, every 1 g reduction in fat intake would require a 2.25 g increase in carbohydrate intake. Up to 10% of total calories of added sugar is allowed, but starches often used to provide bulk and texture in infant and toddler foods are not considered added sugar, and half of grains can be refined grains, all of which could add considerably to a young child’s carbohydrate intake. An “Update to School Nutrition Standards” released in April 2024 by the USDA Food and Nutrition Service still limits milk to fat-free and low-fat but allows flavored milks in school meals so long as the added sugar does not exceed 10% of total calories [140]. In 2020, 29.8% of US children aged 2 to 4 in the Women’s, Infant’s, and Children’s (WIC) supplemental food program for low-income families, which adheres to the *DGA*, were overweight or obese [141]. The number of children with autism disorder increased from one in one hundred and fifty in 1992 to one in thirty-six in 2020 [142]. These alarming trends strongly suggest that the *DGA* recommendations for children, such as consuming only low-fat and fat-free dairy, are problematic.

The rates of other health conditions in the US that are diet-related have also increased significantly. NHANES surveys reported that the prevalence of diabetes in the US, which was estimated at 6.2% in 1994, increased to 9.9% in 2010 [143] and to 14.7% in 2021 [144], and the number of people with Alzheimer’s disease rose from 4.5 million in 2000 [145] to 6.7 million in 2023 [146]. The prevalence of type 2 diabetes has also been increasing steadily throughout the world [147]. Paradoxically, and until recently, people with diabetes were routinely advised to consume a low-fat diet in keeping with guidelines promoted by the lipid–heart hypothesis. To maintain the same level of daily energy intake, a low-fat diet is, by default, a high-carbohydrate diet. Diabetes is a major risk factor for heart disease, dementia, kidney failure, blindness, amputation, and other serious complications, but the trajectory of progression to disability could be altered by adopting a healthy whole-food diet that is lower in carbohydrate and higher in healthy fat. Long-term studies of people with type 2 diabetes, prediabetes, and metabolic syndrome have shown that consuming low-carbohydrate higher-fat diets is safe and can significantly reduce fasting blood glucose, fasting insulin levels, and hemoglobin A1C, reduce the requirement for insulin and other diabetes medications, promote weight loss, reduce systolic and diastolic blood pressure, and improve biomarkers of cardiovascular disease without adverse effects [148,149,150,151]. Low-carbohydrate diets have also been shown to improve glucose control and reduce the insulin requirement in people with type 1 diabetes, and low-carbohydrate diets were often used to treat diabetes prior to the discovery of insulin [152,153].

In 2019, a consensus statement on nutrition therapy for adults with diabetes funded by the American Diabetes Association reviewed low- and very-low-carbohydrate diets and concluded that replacing high-carbohydrate foods with lower-carbohydrate, higher-fat foods may improve glucose control, serum triglycerides, and HDL-C levels [154]. A more recent expert panel on diet and insulin resistance has recommended that the DGA reconsider the current Acceptable Macronutrient Distribution Range (AMDR) for carbohydrate of 45 to 65% because it does not align with the cumulative evidence on low-carbohydrate diets and health outcomes showing that carbohydrate intakes of less than 130 g daily can have a positive impact on body weight and other cardiovascular risk factors, and diets with less than 50 g of carbohydrate could be therapeutic for some metabolic disorders like type 2 diabetes and metabolic syndrome. Glucose can be produced endogenously as needed by several different mechanisms, and therefore, there is no essential requirement for dietary carbohydrate [155]. The consensus of the Volek et al. panel is in alignment with the recommendations of 130 g daily of carbohydrate for children and adults aged 1 year and above published in the *Dietary Reference Intakes* (DRIs) by the Institute of Medicine (IOM) in 2006. The IOM section on dietary fat states that “Since there is no defined intake level of total fat at which an adverse effect occurs, an Upper Limit was not set for total fat” [156]. Based on the IOM DRIs for protein of 0.8 g per kg daily and 130 g carbohydrate daily, a 70 kg person on a 2400-kcal diet could consume about 69% of total calories as fat.

### 6.1. The Saturated Fat Content of Some Nutrient-Dense Foods Has Been Exaggerated

For decades, the AHA and DGA have recommended eating lean meats and fat-free or low-fat dairy and greatly limiting the intake of red meats and eggs. Referring to certain nutrient-dense foods such as whole milk, eggs, and meats as “saturated fats” exaggerates the actual SFA content of these foods which also contain significant amounts of MUFA and PUFA. For example, 100 g of whole milk contains 3.25 g of fat and 1.86 g or 20 kcal as saturated fat. This represents just 1% of a 2000-kcal diet. Likewise, a large egg contains more protein (6 g) than fat (5 g), of which just 2 g or 18 kcal is saturated fat. A 100 g piece of beef contains twice as much protein (27 g) as fat (13 g) with 5.22 g or 47 kcal of SFA, about 2.4% of a 2000-kcal diet.

For someone consuming a 2000 kcal diet, the *DGA 2020* recommends replacing foods containing natural saturated fats with 29 g per day of MUFA and PUFA oils, which all contain saturated fat. There is 4 g of saturated fat in 29 g of soybean oil, which would be equivalent to consuming the saturated fat in 215 gm of whole milk or two whole eggs. Eggs, milk, and meat contain an abundance of protein, vitamins, minerals, and other nutrients, including adequate amounts of omega-3 and omega-6 fatty acids, and therefore, liquid oils are not necessary to meet the recommended intakes of essential fatty acids.

One of the unintended consequences of the DGA recommendation limiting eggs is the deprivation of choline. Choline is an important nutrient present in all cell membranes and in the neurotransmitter acetylcholine [157]. Eggs are one of the richest sources of choline with 168 mg per egg, and 100 g of beef provides about 79 mg. A person would need to eat ten slices of wholewheat bread or nine cups of long-grain rice to equal the amount of choline in one egg. Currently, the DGA dietary goals recommend a limit of about 75 g, or less than two eggs per week, for children aged 12 to 23 months, but few foods are more nutrient-dense than eggs for a growing child. The harm of the lipid–heart hypothesis has gone beyond its attack against saturated fat and cholesterol.

### 6.2. The Modern Diet Is Shifting from Whole Foods to Imitation and Ultra-Processed Foods

In the early 1960s, the AHA was able to raise funds from the government and private organization to undertake the NDHS which incentivized more than 30 food manufacturers to produce low-fat, high-carbohydrate, processed, and imitation foods, many of which contained *trans*-fat. A proliferation of such foods on US grocery store shelves followed. In 1995, thirty years after the NDHS ended, the ILSI expert panel on *trans*-fat reported that the Healthy People 2000 goal of 5000 fat-modified foods had been exceeded with more than 5600 such food products on US grocery shelves [40]. The shift away from whole foods, including natural sources of fat, to these fabricated fat-modified foods is likely a major factor in the newer epidemic of metabolic disorders that are affecting all ages and should be addressed more directly in revisions of the *DGA*.

## 7. Conclusions: Implications for Dietary Guidelines

Ancel Keys’ lipid–heart hypothesis states that total serum cholesterol, TC (ΔChol.), can be lowered by following four dietary recommendations: reduce total fat, reduce saturated fat (ΔS), replace saturated fat with polyunsaturated fat, mainly linoleic acid (ΔP), and reduce dietary cholesterol (ΔZ). Keys proposed the following equation based on his hypothesis:ΔChol. = 1.3(2ΔS − ΔP) + 1.5ΔZ.

An analysis of each recommendation reveals the errors of this hypothesis: first, reducing total fat intake was not proven to have any effect on lowering cholesterol levels or dying from heart disease; second, saturated fat, ΔS, included *trans*-fats which raise ΔChol. and assumed that all saturated fatty acids have the same effects on ΔChol.; third, polyunsaturated fat, ΔP, referred mainly to linoleic acid which was recommended without limit, while ignoring oleic acid and alpha-linolenic acid; and fourth, dietary cholesterol (ΔZ) does not account for the complexity of cholesterol metabolism. Evidence from observational and epidemiological studies do not support the lipid–heart hypothesis, and the clinical studies that were designed according to this hypothesis showed that ΔChol. levels could be reduced by replacing saturated fat with polyunsaturated fat but the rate of deaths from heart disease did not change and even increased in some cases. Appendix A presents a brief summary of the history of the lipid–heart hypothesis.

A 1987 *New York Times* article on the cholesterol controversy suggests that Keys had softened his position on cholesterol: “I’ve come to think that cholesterol is not as important as we used to think it was… Let’s reduce cholesterol by reasonable means but let’s not get too excited about it” [158]. Keys made this statement three years after he abandoned his efforts to prove his equation and around the time that the first paper with the partial results of the unsuccessful Minnesota Coronary Experiment was being prepared for the publication in which he was not included as an author. The lipid–heart hypothesis should be abandoned, as Keys himself appeared to be saying in 1987. However, as this historical review has noted, the lipid–heart hypothesis has not only failed to prevent heart disease, but it has also harmed the overall health of people, from children to adults, in so many ways. Using the lipid–heart hypothesis as support, natural diets are being replaced by synthetic diets with dire consequences for global health.

In 1957, the AHA began to promote the low-fat Prudent Diet based on the assumption that this practice would lower the risk of death from heart disease or even reduce serum cholesterol levels, but this idea was already disproven before 1970 in the Seven Countries and the National Diet Heart Studies and by the MRFIT study before the first *DGA* was published in 1980. Likewise, the idea of reducing serum cholesterol levels by replacing natural sources of saturated fat with polyunsaturated fat to lower the risk of cardiac death was disproven in these studies, and concerns about the excessive intake of linoleic acid surfaced in the Minnesota Coronary Experiment and the Sydney Diet Heart Study, as discussed previously in this review. The results of these studies were seriously compromised by the conflation of natural sources of saturated fat and industrial *trans*-fats as solid fat to represent saturated fat. None of these studies reported a significant association of dietary cholesterol intake with serum cholesterol levels or cardiac mortality. Yet, decades later, the public is still advised by the *DGA 2020–2025* to consume a low-fat diet of 20 to 35% of total calories, to limit intake of foods containing natural saturated fat, to limit saturated fat intake to less than 10%, and to consume as little dietary cholesterol as possible. In addition, the *DGA 2020–2025* continues to promote the consumption of unsaturated high-linoleic oils, presumably to lower serum cholesterol levels, but at the expense of natural nutrient-dense sources of fat, like whole-fat dairy, eggs, and meat, for which the actual saturated fat content has been exaggerated, as discussed in Section 6.1.

The *DGA 2025–2030* Advisory Committee should strongly reconsider the perpetuation of the low-fat, low-saturated-fat, high-PUFA guidelines, which may be especially detrimental to growing children and people with prediabetes and diabetes. Instead, the Committee should formulate dietary patterns comprising whole foods that address the diversity of cultures in the US and beyond and should discontinue the recommendations to consume only fat-free or low-fat dairy and to limit the consumption of cholesterol-rich but nutrient-dense foods like eggs and meat which contain not only SFA but also MUFA and PUFA and are rich in choline, vitamins, and minerals. The *DGA 2020–2025* recommends consuming a variety of proteins, vegetables, and fruits and could certainly recommend consuming fats and oils from a variety of sources to achieve a balance of SFA, MUFA, and PUFA. The Committee should discourage the consumption of imitation and ultra-processed foods, heavily hydrogenated fats, which are still allowed by the FDA, excessive amounts of high-linoleic oils, and reheated fats and oils, which accumulate oxidized byproducts. The Committee should also consider including starches used for bulk and texture as “added sugar”, particularly in foods for infants and very young children, and encourage the consumption of whole grains, rather than setting a limit that allows 50% of grains to be refined, which are depleted of fiber and other important nutrients. Finally, the *DGA 2025–2030* Advisory Committee should review the body of evidence on the benefits of low-carbohydrate and very-low-carbohydrate diets for people with insulin resistance disorders and reassess the AMDR for carbohydrate. In summary, the *DGA 2025–2030* should reconsider its support for the lipid–heart hypothesis and promote the shift of the American diet back to traditional whole foods and away from fabricated fat-modified foods to overcome the current epidemic of metabolic disorders, which would in turn reduce the prevalence of cardiovascular disease.

The lipid–heart hypothesis should not be used in the formulation of dietary guidelines.

## Figures and Tables

**Figure 1 nutrients-16-01447-f001:**
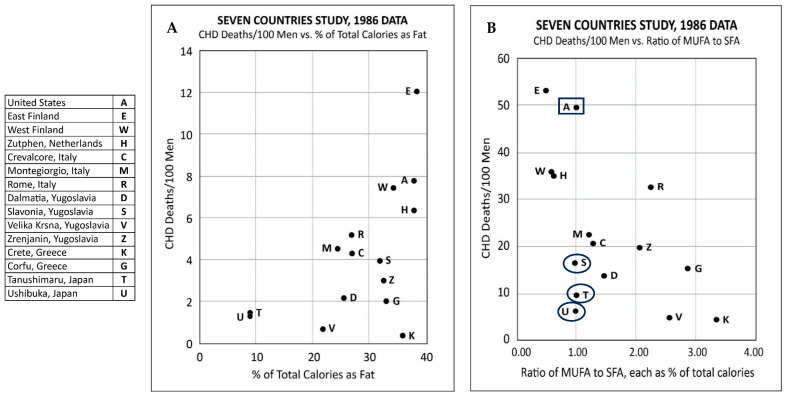
Results from the Seven Countries Study. (Figure 1**A**) There was no association between the % of total calories as fat and CHD deaths. Crete (**K**) had the lowest all-cause and CHD deaths but had one of the highest fat intakes at 36.1%. However, the East Finland (E) cohort consumed a comparable amount of fat at 38.5% and had the highest number of CHD deaths. (Figure 1**B**) Keys claimed that there was an association between CHD deaths and the ratio of (MUFA/SFA) intake. However, the fifteen-year analysis of data showed otherwise: three cohorts with the lowest CHD death rates (Tanushimaru (**T**), Ushibuka (**U**), and Slavonia (**S**)) had the same MUFA/SFA ratio of 1.0 as the cohort which had the second highest CHD deaths (US railroad men (**A**)). (The legends are those used in [24]).

**Figure 2 nutrients-16-01447-f002:**
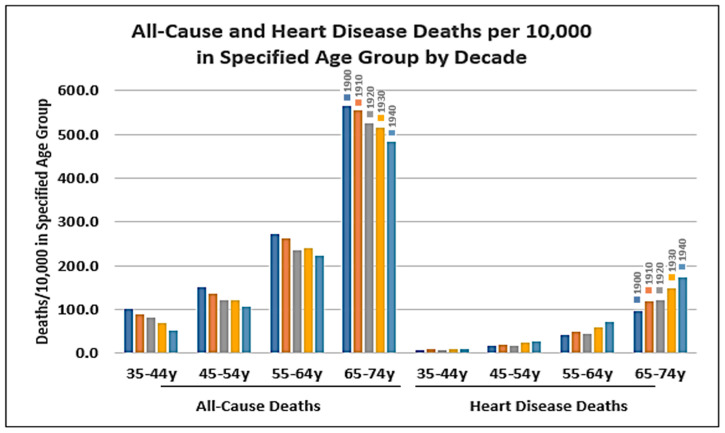
In 1900, infections were the top three causes of death. Between 1900 and 1940, public health infection control measures led to dramatic reductions in all-cause deaths in all age groups, and life expectancy steadily increased. Fewer deaths were attributed to heart disease in infants, children, and young adults, and many more people survived to middle and old age with proportionately more deaths attributed to heart disease rather than infection. Thus, there appeared to be an epidemic of heart disease in middle-aged and older men. However, fewer middle-aged and older men were dying prematurely, and most were dying from causes other than heart disease. (y = years) [108].

**Figure 3 nutrients-16-01447-f003:**
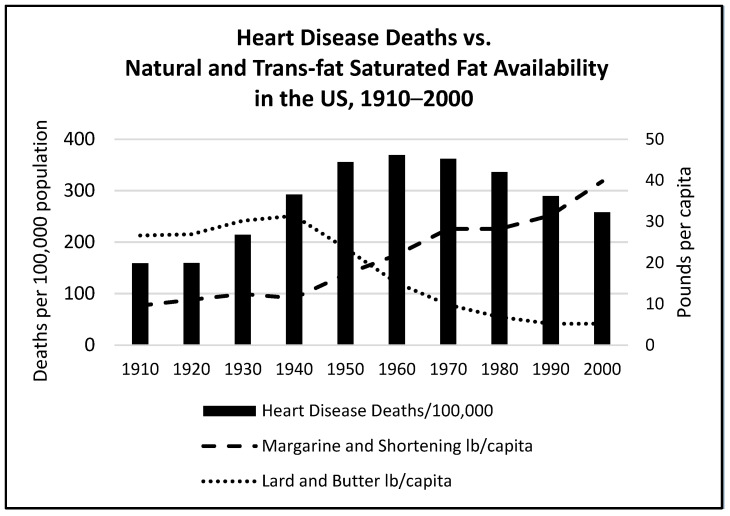
From 1910 to 2000, the availability of butter and lard declined while industrial *trans*-fats (margarine and shortening) increased dramatically. Over the same period, deaths from heart disease also escalated. This suggests that, if fat was a factor, *trans*-fat was more likely responsible for the increase in heart disease than butter and lard. Abbreviation: lb = pounds. Source: USDA ERS Data on Added Fats from 1909 to 2017. https://view.officeapps.live.com/op/view.aspx?src=https%3A%2F%2Fwww.ers.usda.gov%2Fwebdocs%2FDataFiles%2F50472%2Ffats.xls%3Fv%3D3307.7&wdOrigin=BROWSELINK, accessed on 4 March 2024.

**Figure 4 nutrients-16-01447-f004:**
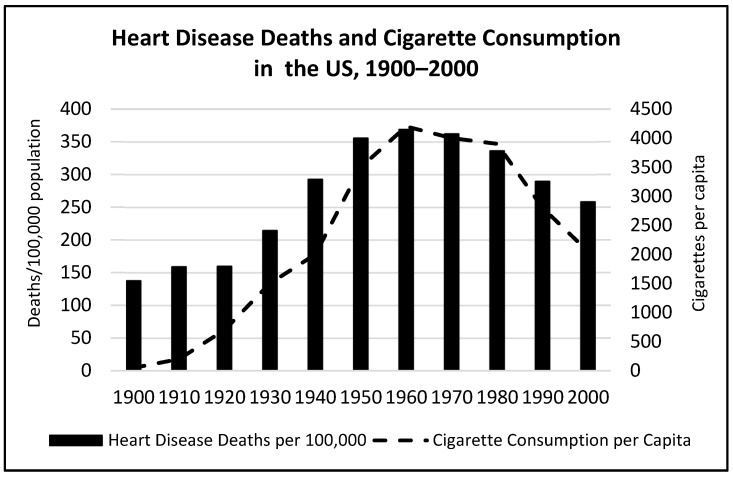
There was a parallel decrease in tobacco smoking and heart disease in the US from 1960 onwards. A CDC report on public health advances considered the decline in tobacco an important factor and did not mention changes in dietary fat consumption as a factor [113].

**Figure 5 nutrients-16-01447-f005:**
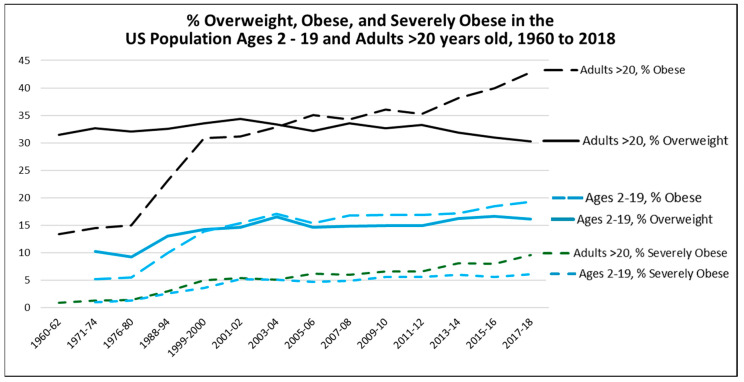
Since the institution of the low-fat, low-saturated fat Dietary Guidelines for Americans in 1980, the prevalence of obesity in US adults, children, and adolescents has more than tripled, and extreme obesity has increased 10-fold in adults and 6-fold in children and adolescents. Body mass index (BMI) is defined as follows (in kg/m^2^): for adults, overweight: 25.0–29.9; obese: ≥30.0; and severely obese: ≥40.0. For children, BMI is defined by percentile: overweight is above the 85th percentile and below the 95th percentile; obese is at or above the 95th percentile; severely obese is at or above 120% of the 95th percentile. Data sources: 1. National Center for Health Statistics, National Health Examination Survey, 1960–1962, and National Health and Nutrition Examination Surveys, 1971–1974, 1976–1980, 1988–1994, and 1999–2018. 2. Fryar CD, Carroll MD, Afful J. Prevalence of overweight, obesity, and severe obesity among adults aged 20 and over: United States, 1960–1962 through 2017–2018. NCHS Health E-Stats. 2020. 3. Fryar CD, Carroll MD, Afful J. Prevalence of overweight, obesity, and severe obesity among children and adolescents aged 2–19 years: United States, 1963–1965 through 2017–2018. NCHS Health E-Stats. 2020.

## Data Availability

All data are available in the cited scientific articles, books, and websites.

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
