# Peer review of "The Lipid–Heart Hypothesis and the Keys Equation Defined the Dietary Guidelines but Ignored the Impact of Trans-Fat and High Linoleic Acid Consumption"

_nutrients, 2024, doi:10.3390/nu16101447_

Round 1
Reviewer 1 Report
Comments and Suggestions for Authors
Overall, I believe that the manuscript presented for review is a very valuable and interesting presentation of the issue of lipid-heart hypothesis over the years.
I have some minor comments:
- references in the text are not cited according to journal guidelines
- references should be checked carefully, e.g. ref. 129 and 130 - the same item; ref. 158 - four separate references
- references are missing in some fragments, e.g. lines 607-613, 850-862
- lines 30, 209, 242 - it looks like sth was missing at the beginning of the subtitle
- lines 172-179 - this fragments seems to be out of the place, consider to rewrite or move it
In general, in some parts my impression is that the authors approach to the subject quite emotional, and subjective (?), e.g. point 5.6 the authors claim that MCE study was designed and led by A. Keys, and that some results were published by Frantz in 1989, excluding Keys as co-author (lines 811-813) - is it fact or own assumption or feeling of the authors?
Author Response
Nutrients Manuscript 2981637
Reviewer 1 Comments and Author Responses:
Comment 1. Overall, I believe that the manuscript presented for review is a very valuable and interesting presentation of the issue of lipid-heart hypothesis over the years.
Response: Thank you very much for this encouraging comment and your scoring of the paper. We also appreciate your keen eye and attention to detail in your subsequent comments.
Comment 2: references in the text are not cited according to journal guidelines
Response: We apologize for missing this important detail in the journal guidelines and have changed the superscripts to brackets.
Comment 3: references should be checked carefully, e.g. ref. 129 and 130 - the same item; ref. 158 - four separate references
Response: Thank you for noting these issues with the duplicate reference.
Regarding the duplicate references #129 and 130, the missing reference has been added and the two references are now # 133 and 134 due to the addition of new references preceding them. The additional reference is now:
- Smoking and Health: Report of the Advisory Committee to the Surgeon General of the Public Health Service. United States. Public Health Service. Office of the Surgeon General, 1964. Available at: https://profiles.nlm.nih.gov/spotlight/nn/catalog/nlm:nlmuid-101584932X202-doc
Regarding reference 158 (now reference 162) which listed four separate references by Dayton et al., we have decided to keep only the 1969 paper since all information in the other three references are earlier reports of the same study and are discussed in detail and cited in the 1969 paper:
- Dayton S, ML Pearce, S Hashimoto, WJ Dixon, U Tomiyasu. A controlled clinical trial of a diet high in unsaturated fat in preventing complications of atherosclerosis. Circulation 1969; 40(Suppl. II): II1-63.
Also, related to this change in the reference, in line 913, we have revised: “Dayton and co-workers in five papers reported…” to read “Dayton and co-workers reported…”.
In addition, the next sentence at line 916 was expanded to read:
“They also found that there were no differences between the high-PUFA diet and control groups in the percentages of triglycerides, free fatty acids, cholesterol, cholesterol esters, or phosphatides in the aorta, coronary arteries, or atheromatous plaques or in the amount of calcium in the aorta.”
Comment 4: references are missing in some fragments, e.g. lines 607-613, 850-862
Response: We agree with your observations. Regarding lines 607 to 613 (now lines 674-676), which is the introduction to a lengthy section, we have removed this paragraph and replaced it with the following sentence which does not hint at the results and therefore would not require references:
“This section will discuss several important large long-term interventional clinical trials and autopsy studies of dietary fats and the lipid-heart hypothesis which were conducted between 1963 and 1980.”
Regarding lines 850 to 862, now lines 907-928, we have added references for each of the bullet points, have added clarifying language to the first bullet point, and have removed one bullet point about the percentage of fat in the diet, and simplified the summary sentence beginning in line 927. Upon review of the autopsy studies, the investigators did not appear to specifically address differences in atherosclerosis with lower versus higher fat diets. The findings on low versus high fat are related to cardiac mortality and imaging studies but not specifically to autopsy findings. This fragment will now read as follows:
“Further supporting the autopsy findings reported in the MCE [[i]], Dayton and co-workers [[ii]] reported no differences in autopsy studies in the degree of atherosclerosis or numbers of atheromatous plaques in the men consuming a high-PUFA diet versus control diet. They also found that there were no differences between the high-PUFA diet and control groups in the percentages of triglycerides, free fatty acids, cholesterol, cholesterol esters, or phosphatides in the aorta, coronary arteries, or atheromatous plaques or in the amount of calcium in the aorta. In summary, human autopsy studies showed that the degree of atherosclerosis and atheroma formation are the same:
- In people who had evidence of myocardial infarctions and those who do not [161].
- In people with high versus low serum cholesterol levels [159].
- In people who eat higher versus lower percentages of energy as polyunsaturated fat [152,161].
- In people who replace saturated fat with polyunsaturated fat [152,161].
Thus, autopsy findings have failed to confirm that replacing saturated fat with polyunsaturated fat reduces the degree of atherosclerosis or atheroma formation.”
Comment 5: lines 30, 209, 242 - it looks like sth was missing at the beginning of the subtitle
Response: You are correct. In the manuscript we submitted, there were symbols for “delta” (Δ) in each of those headings, specifically this should be “ΔS” at line 30 (now line 66), “ΔP” at line 209 (now line 260), and “ΔZ” at line 242 (now line 294), but it appears that a squiggle and a period replaced the delta symbol. We have added the delta symbols in the three headings.
Comment 6: lines 172-179 - this fragment seems to be out of the place, consider to rewrite or move it
Response: Thank you for noticing that this fragment seems out of place, and we agree. We have moved the fragment from lines 172-179 to line 361-369 to expand on a paragraph about the importance of endogenous cholesterol and saturated fat. We also revised the information on cholesterol beginning at line 354 as follows:
Instead of “Cholesterol is endogenously produced in all human cells as needed for numerous physiological processes and is a precursor for hormones and many other substances…” this information will now read: “Cholesterol is endogenously produced in all human cells as needed for numerous physiological processes. Cholesterol provides structural support within the membranes of cells and organelles as well as for the delicate neuronal networks in the brain and spinal cord. Cholesterol is also a signaling molecule and a precursor for vitamins D, hormones, bile salts, and other substances.”
Comment 7: In general, in some parts my impression is that the authors approach to the subject quite emotional, and subjective (?), e.g. point 5.6 the authors claim that MCE study was designed and led by A. Keys, and that some results were published by Frantz in 1989, excluding Keys as co-author (lines 811-813) - is it fact or own assumption or feeling of the authors?
Response: Thank you for your insightful observation. We checked with several sources including direct email communication with Christopher Ramsden, whose 2016 article on the MCE study is discussed cited in the manuscript. We have made the following changes to provide clarification as follows:
In lines 863 to 865, we changed this statement: “The Minnesota Coronary Experiment (MCE) (US, 1968 to 1973) was another study designed and led by Ancel Keys which was expected …” to “The Minnesota Coronary Experiment (MCE) (US, 1968 to 1973), which was led by co-principal investigators Ancel Keys and Ivan Frantz, was expected to demonstrate the benefit of a high PUFA diet.”
At line 878, we have removed: “although Keys was the principal investigator”.
_________________________
Thank you again for your kind review and your appropriate and helpful suggestions.
Best regards,
Mary Newport and Fabian Dayrit
[i]. [161] Ramsden CE, Zamora D, Majchrzak-Hong S, Faurot KR, Broste SK, Frantz RP, Davis JM, Ringel A, Suchindran CM, Hibbeln JR. Re-evaluation of the traditional diet-heart hypothesis: analysis of recovered data from Minnesota Coronary Experiment (1968-73). BMJ 2016; 353: i1246.
[ii]. [162] Dayton S, ML Pearce, S Hashimoto, WJ Dixon, U Tomiyasu. A controlled clinical trial of a diet high in unsaturated fat in preventing complications of atherosclerosis. Circulation 1969; 40(Suppl. II): II1-63.
Reviewer 2 Report
Comments and Suggestions for Authors
Dear Authors,
This is a very well-written and interesting review. It is rare that a reviewer does not have many comments, but this work appears to be complete after some minor modifications and adding citations.
The article is clear, relevant, and presented in a well-structured manner. Your review provides a critical evaluation of Ancel Keys' lipid-heart hypothesis and its implications for dietary guidelines related to heart disease. Here are some key points and suggestions for refining your review:
1 . To enhance the review, consider further elaborating on the specific limitations and consequences of conflating saturated fats with trans-fats. How did this oversight affect subsequent research directions and public health policies? Providing a critical evaluation can deepen the discussion.
2. Are there other critical evaluations from 2022-2024. It would be good to refer to them as well and compare your opinions.
3. In the summary, it would be worth adding what can be done to modify the existing guidelines.
Kind regards,
Reviewer
Author Response
Nutrients 2981637 - Author Responses to Reviewer 2
Reviewer 2 Comments and Suggestions for Authors and the Authors' Responses:
Comment 1. Dear Authors,
This is a very well-written and interesting review. It is rare that a reviewer does not have many comments, but this work appears to be complete after some minor modifications and adding citations.
The article is clear, relevant, and presented in a well-structured manner. Your review provides a critical evaluation of Ancel Keys' lipid-heart hypothesis and its implications for dietary guidelines related to heart disease. Here are some key points and suggestions for refining your review:
To enhance the review, consider further elaborating on the specific limitations and consequences of conflating saturated fats with trans-fats. How did this oversight affect subsequent research directions and public health policies? Providing a critical evaluation can deepen the discussion.
Response: Thank you so much for these very kind comments, your scoring of the paper, and your insightful and helpful suggestions. We greatly appreciate the opportunity to expand our discussion on public health policies related to nutrition. The paragraph on saturated fat beginning at line 172 has moved to line 361. To elaborate on the limitations and consequences of the conflation of saturated fats with trans-fats, we have added the following statement as a new paragraph beginning in line 214:
“From 1961, when the AHA published its first advisory until 2006, when the FDA mandated the labeling of trans-fats, saturated fat and trans-fats were not separately labeled, making dietary studies on saturated fats inherently unreliable. The 1961 AHA advice to reduce total fat intake and avoid foods containing natural saturated fat, such as whole fat dairy, eggs, and other animal fat, relied on animal studies and small metabolic ward studies and was premature, occurring before the first large scale study in people had taken place. This guidance has not changed in the DGA despite a lack of evidence that heeding this advice will reduce a person’s risk of dying from heart disease. Furthermore, this guidance has promoted a persistent, unwarranted fear of consuming nutrient dense foods which have been consumed by humans for many millennia, such as meat, eggs, dairy, and coconut. As a result, since the mid-twentieth century, the diets in the US have gradually shifted away from traditional whole foods toward imitation and ultra-processed foods which often contain high-fructose corn syrup, refined grains, synthetic vitamins, preservatives, and other additives, which are not nutritionally equivalent to whole foods. As a result, the US has experienced increasing rates of metabolic disorders in people of all ages, and it is reasonable to believe that this new epidemic is related to this major shift in dietary patterns. This will be discussed in greater detail below.”
Comment 2. Are there other critical evaluations from 2022-2024. It would be good to refer to them as well and compare your opinions.
Response: Thank you for this suggestion. Beginning after the words, “lack of science behind the lipid-heart hypothesis” in line 40, we will move reference 4 and insert the following statement and the related additional references as noted, which will be properly formatted in the manuscript:
“Recent reviews have addressed the lack of evidence that saturated fat in general or in specific foods, such as milk and eggs, causes cardiovascular disease (CVD) or that reducing saturated fat intake lowers CVD risk, the importance of LDL-C particle size and distribution pattern in CVD rather than level of total LDL-C, and the importance of the food matrix and overall dietary pattern which affect digestion, absorption, and other properties of specific nutrients [[i],[ii],[iii]]. Other critiques have focused on exposing reliance on insufficient evidence [[iv]], food industry pressure [[v]], and biases and conflicts of interest [[vi],[vii]] in the formulation of dietary guidelines.”
Comment 3. In the summary, it would be worth adding what can be done to modify the existing guidelines.
Response: We agree and greatly appreciate the opportunity to expand the discussion on what can be done specifically to modify the existing guidelines. As a prelude to the discussion within the conclusion section, we have added new information in Sections 6 and 7 with additional references.
New information has been added beginning at line 946:
“It is reasonable to assume that human milk is designed to promote optimal growth and development of the brain and other organs in the rapidly growing infant and young child. A study of 64 traditional cultures conducted in the 1950s reported breastfeeding in some populations up to age 4 or 5, with an average age of weaning estimated at 2.8 years [[viii]]. However, the CDC reported that, in 2019, by six months of age, just 55.8% of infants received any breastmilk and only 24.9% were exclusively breastfed, and therefore, most US infants receive substantial infant formula during the first 1 to 2 years of life [[ix]]. USDA FoodData Central reports that human milk contains about 56% of total calories as fat, including 26% of total calories as saturated fat, 15% as MUFA, and 4.5% as PUFA. By comparison, the three leading infant formulas in the US contain on average 47.5% of calories as fat, with 18.7% as saturated fat, but more PUFA (9.8%) than human milk. Infant formulas contain considerably more linoleic acid (8.6%) than human milk (4.8%). It is noteworthy that most commercial infant formulas contain coconut oil to provide the medium-chain fatty acids and myristic acid which are found in human milk but not found in most other common vegetable oils [[x]].
“DGA 2020 advised an abrupt reduction in total fat to 30-40% and saturated fat intake to <10% of total calories when a child turns two years old at a time when the brain and body are rapidly growing and developing. Evidence to support this guidance in long-term clinical trials is lacking. The USDA’s My Plate for Preschoolers, which is based on DGA 2020 shows no fat on the plate and advises that even very young children eat only fat-free or low-fat dairy and lean meats [[xi]]. This gives parents the impression that fat is unhealthy for children. With the protein requirement remaining constant, to provide adequate calories for growth every 1 g reduction in fat intake would require a 2.25 g increase in carbohydrate intake. Up to 10% of total calories of added sugar is allowed, but starches often used to provide bulk and texture in infant and toddler foods are not considered added sugar, and half of grains can be refined grains, all of which could add considerably to a young child’s carbohydrate intake. An “Update to School Nutrition Standards” released in April 2024 by the USDA Food and Nutrition Service still limits milk to fat-free and low-fat but allows flavored milks in school meals so long as the added sugar does not exceed 10% of total calories [[xii]]. In 2020, 29.8 % of US children ages 2 to 4 in the Women’s, Infant’s, and Children’s (WIC) supplemental food program for low-income families, which adheres to the DGA, were overweight or obese [[xiii]]. The number of children with autism disorder increased from one in 150 in 1992 to one in 36 in 2020 [[xiv]].”
In line 970, the highlighted sentence is included here rather than later in the discussion:
“In 2020, 29.8 % of US children ages 2 to 4 in the Women’s, Infant’s, and Children’s (WIC) supplemental food program for low-income families, which adheres to the DGA, were overweight or obese [[xv]]. In addition, the number of children with autism spectrum disorders increased from one in 150 in 1992 to one in 36 in 2020 [[xvi]]. These alarming trends strongly suggest that the DGA recommendations for children, such as consuming only low-fat and fat-free dairy, are problematic.”
Also, as a prelude to discussion on how to change the DGA, new information related to the epidemic of metabolic disorders is included in Section 6 beginning on line 985:
“The prevalence of type 2 diabetes has also been increasing steadily throughout the world [[xvii]]. Paradoxically, and until recently, people with diabetes were routinely advised to consume a low-fat diet in keeping with guidelines promoted by the lipid-heart hypothesis. To maintain the same level of daily energy intake, a low-fat diet is, by default, a high-carbohydrate diet. Diabetes is a major risk factor for heart disease, dementia, kidney failure, blindness, amputation, and other serious complications but the trajectory of progression to disability could be altered by adopting a healthy whole food diet that is lower in carbohydrate and higher in healthy fat. Long-term studies of people with type 2 diabetes, prediabetes, and metabolic syndrome have shown that consuming low-carbohydrate higher-fat diets are safe and can significantly reduce fasting blood glucose. fasting insulin levels, and hemoglobin A1C, reduce the requirement for insulin and other diabetes medications, promote weight loss, reduce systolic and diastolic blood pressure, and improve biomarkers of cardiovascular disease without adverse effects [[xviii],[xix],[xx],[xxi]]. Low-carbohydrate diets have also been shown to improve glucose control and reduce the insulin requirement in people with type 1 diabetes and low-carbohydrate diets were often used to treat diabetes prior to the discovery of insulin [[xxii],[xxiii]].
“In 2019, a consensus statement on nutrition therapy for adults with diabetes funded by the American Diabetes Association reviewed low and very-low carbohydrate diets and concluded that replacing high carbohydrate foods with lower carbohydrate, higher fat foods may improve glucose control, serum triglycerides, and HDL-C levels [[xxiv]]. A more recent expert panel on diet and insulin resistance has recommended that the DGA reconsider the current Acceptable Macronutrient Distribution Range (AMDR) for carbohydrate of 45 to 65% because it does not align with the cumulative evidence on low-carbohydrate diets and health outcomes showing that carbohydrate intakes of less than 130 g daily can have a positive impact on body weight and other cardiovascular risk factors, and diets with fewer than 50 g carbohydrate could be therapeutic for some metabolic disorders like type 2 diabetes and metabolic syndrome. Glucose can be produced endogenously as needed by several different mechanisms, and therefore, there is no essential requirement for dietary carbohydrate [[xxv]]. The consensus of the Volek et al. panel is in alignment with the recommendations of 130 g daily of carbohydrate for children and adults ages 1 year and above published in the Dietary Reference Intakes (DRI) by the Institute of Medicine (IOM) in 2006. The IOM section on dietary fat states that, “Since there is no defined intake level of total fat at which an adverse effect occurs, an Upper Limit was not set for total fat” [[xxvi]]. Based on the IOM DRI for protein of 0.8 g per kg daily and 130 g carbohydrate daily, a 70-kg person on a 2400-kcal diet could consume about 69% of total calories as fat.”
A new section with additional information as a prelude to our recommendation for changes to the DGA has been added beginning at line 1023:
“6.1 The Saturated Fat Content of Some Nutrient-Dense Foods Has Been Exaggerated
“For decades the AHA and DGA have recommended eating lean meats, fat-free or low-fat dairy, and greatly limiting intake of red meats and eggs. Referring to certain nutrient-dense foods such as whole milk, eggs, and meats as “saturated fats” exaggerates the actual SFA content of these foods which also contain significant amounts of MUFA and PUFA. For example, 100 grams of whole milk contains 3.25 g of fat and 1.86 g or 20 kcal as saturated fat. This represents just 1% of a 2000-kcal diet. Likewise, a large egg contains more protein (6 g) than fat (5 g), of which just 2 g or 18 kcal is saturated fat. A 100 g piece of beef contains twice as much protein (27 g) as fat (13 g) with 5.22 g or 47 kcal of SFA, about 2.4% of a 2000-kcal diet.
“For someone consuming a 2000 kcal diet, DGA 2020 recommends replacing foods containing natural saturated fats with 29 g per day of MUFA and PUFA oils, which all contain saturated fat. There are 4 g of saturated fat in 29 g of soybean oil which would be equivalent to consuming the saturated fat in 215 gm of whole milk or two whole eggs. Eggs, milk, and meat contain an abundance of protein, vitamins, minerals, and other nutrients, including adequate amounts of omega-3 and omega-6 fatty acids, and therefore, liquid oils are not necessary to meet the recommended intakes of essential fatty acids.
“One of the unintended consequences of the DGA recommendation limiting eggs is deprivation of choline. Choline is an important nutrient present in all cell membranes and in the neurotransmitter acetylcholine [[xxvii]]. Eggs are one of the richest sources of choline with 168 mg per egg, and 100 g of beef provides about 79 mg. A person would need to eat ten slices of whole wheat bread or 9 cups of long grain rice to equal the amount of choline in one egg. Currently, the DGA dietary goals recommend a limit of about 75 grams, or less than two eggs per week, for children ages 12 to 23 months, but few foods are more nutrient dense than eggs for a growing child. The harm of the lipid-heart hypothesis has gone beyond its attack against saturated fat and cholesterol.”
A new heading has been added to separate the section that follows from the previous information:
“6.2 The Modern Diet Is Shifting from Whole Foods to Imitation and Ultra-Processed Foods”
In the conclusion section, we have added the following:
“However, as this historical review has noted, the lipid-heart hypothesis has not only failed to prevent heart disease, it has also harmed the overall health of people, from children to adults, in so many ways. Using the lipid-heart hypothesis as support, natural diets are being replaced by synthetic diets with dire consequences for global health.
“In 1957, the AHA began to promote the low-fat Prudent Diet based on the assumption that this practice would lower the risk of death from heart disease or even reduce serum cholesterol levels, but this idea was already disproven before 1970 in the Seven Countries and the National Diet Heart Studies, and by the MRFIT study before the first DGA was published in 1980. Likewise, the idea of reducing serum cholesterol levels by replacing natural sources of saturated fat with polyunsaturated fat to lower risk of cardiac death was disproven in these studies and concerns about excessive intake of linoleic acid surfaced in the Minnesota Coronary Experiment and the Sydney Diet Heart Study, as discussed previously in this review. The results of these studies were seriously compromised by the conflation of natural sources of saturated fat and industrial trans-fats as solid fat to represent saturated fat. None of these studies reported a significant association of dietary cholesterol intake with serum cholesterol levels or cardiac mortality. Yet, decades later, the public is still advised by DGA 2020-2025 to consume a low-fat diet of 20 to 35% of total calories, to limit intake of foods containing natural saturated fat, to limit saturated fat intake to less than 10%, and to consume as little dietary cholesterol as possible. In addition, DGA 2020-2025 continues to promote consumption of unsaturated high-linoleic oils, presumably to lower serum cholesterol levels, but at the expense of natural nutrient-dense sources of fat, like whole fat dairy, eggs, and meat, for which the actual saturated fat content has been exaggerated, as discussed in section 6.1.
“The DGA 2025-2030 Advisory Committee should strongly reconsider perpetuation of the low-fat, low-saturated fat, high-PUFA guidelines, which may be especially detrimental to growing children and people with prediabetes and diabetes. Instead, the Committee should formulate dietary patterns comprised of whole foods that address the diversity of cultures in the US and beyond and should discontinue the recommendations to consume only fat-free or low-fat dairy and to limit consumption of cholesterol-rich but nutrient dense foods like eggs and meat which contain not only SFA but also MUFA and PUFA and are rich in choline, vitamins, and minerals. DGA 2020-2025 recommends consuming a variety of proteins, vegetables, and fruits and could certainly recommend consuming fats and oils from a variety of sources to achieve a balance of SFA, MUFA, and PUFA. The Committee should discourage consumption of imitation and ultra-processed foods, heavily hydrogenated fats, which are still allowed by the FDA, excessive amounts of high-linoleic oils, and reheated fats and oils, which accumulate oxidized byproducts. The Committee should also consider including starches used for bulk and texture as “added sugar”, particularly in foods for infants and very young children, and encourage consumption of whole grains, rather than setting a limit that allows 50% of grains to be refined, which are depleted of fiber and other important nutrients. Finally, the DGA 2025-2030 Advisory Committee should review the body of evidence on the benefits of low-carbohydrate and very-low carbohydrate diets to people with insulin resistance disorders and reassess the AMDR for carbohydrate. In summary, DGA 2025-2030 should reconsider its support for the lipid-heart hypothesis and promote the shift of the American diet back to traditional whole foods and away from fabricated fat-modified foods to overcome the current epidemic of metabolic disorders, which would in turn reduce the prevalence of cardiovascular disease.
“The lipid-heart hypothesis should not be used in the formulation of dietary guidelines.”
_______________
The additional references are shown below. Thank you very much for your kind and helpful review and suggestions. We are grateful for the opportunity to add significantly to the discussion on the issues in the Dietary Guidelines for Americans!
Best regards,
Mary Newport and Fabian Dayrit
[i]. Astrup A, Magkos F, Bier DM, Brenna JT, de Oliveira Otto MC, Hill JO, King JC, Mente A, Ordovas JM, Volek JS, Yusuf S, Krauss RM. Saturated fats and health: A reassessment and proposal for food-based recommendations: JACC state-of-the-art review. J Am Coll Cardiol 2020; 76(7): 844-857.
[ii]. Astrup A, Teicholz N, Magkos F, Bier DM, Brenna JT, King JC, Mente A, Ordovas JM, Volek JS, Yusuf S, Krauss RM. Dietary saturated fats and health: Are the U.S. guidelines evidence-based? Nutrients 2021; 13(10): 3305.
[iii]. Hirahatake KM, Astrup A, Hill JO, Slavin JL, Allison DB, Maki KC. Potential cardiometabolic health benefits of full-fat dairy: The evidence base. Adv Nutr 2020; 11(3): 533-547.
[iv]. Achterberg C, Astrup A, Bier DM, King JC, Krauss RM, Teicholz N, Volek JS. An analysis of the recent US dietary guidelines process in light of its federal mandate and a National Academies report. PNAS Nexus 2022; 1(3): pgac107.
[v]. Kearns CE, Schmidt LA, Glantz SA. Sugar industry and coronary heart disease research: A historical analysis of internal industry documents. JAMA Intern Med 2016; 176(11): 1680-1685.
[vi]. Mialon M, Serodio P, Crosbie E, Teicholz N, Naik A, Carriedo A. Conflicts of interest for Dietary Guidelines Advisory Committee members: Neither a new nor unexplored issue. Adv Nutr 2023; 14(5):1246-1247.
[vii]. Teicholz N. A short history of saturated fat: the making and unmaking of a scientific consensus. Curr Opin Endocrinol Diabetes Obes 2023; 30(1):65-71.
[viii]. Stuart-Macadam P, Dettwyler K. Breastfeeding: Biocultural Perspectives. Aldine Transaction (1995).
[ix]. CDC. Breastfeeding Report Card, 2022. Available at: https://www.cdc.gov/breastfeeding/data/reportcard.htm.
[x]. USDA FoodData Central. United States Department of Agriculture. Available at: https://fdc.nal.usda.gov.
[xi]. USDA MyPlate Nutrition Information for Preschoolers. United States Department of Agriculture. Available at: https://www.myplate.gov/life-stages/preschoolers
[xii]. Updates to the School Nutrition Standards | Food and Nutrition Service (usda.gov) USDA Food and Nutrition Services. Updates to the School Nutrition Standards, 2024. Available at: https://www.fns.usda.gov/cn/school-nutrition-standards-updates
[xiii]. Obesity Among Young Children Enrolled in WIC: Overweight & Obesity. Table 2. https://www.cdc.gov/obesity/data/obesity-among-WIC-enrolled-young-children.html%23trends
[xiv]. Data & Statistics on Autism Spectrum Disorder. CDC. https://www.cdc.gov/ncbddd/autism/data.html
[xv]. Obesity Among Young Children Enrolled in WIC: Overweight & Obesity. Table 2. https://www.cdc.gov/obesity/data/obesity-among-WIC-enrolled-young-children.html%23trends
[xvi]. Data & Statistics on Autism Spectrum Disorder. CDC. https://www.cdc.gov/ncbddd/autism/data.html
[xvii]. International Diabetes Federation Diabetes Atlas, 10th Edition, 2021. Available at: https://diabetesatlas.org/idfawp/resource-«les/2021/07/IDF_Atlas_10th _Edition_2021.pdf.
[xviii]. Volek JS, Phinney SD, Forsythe CE, Quann EE, Wood RJ, Puglisi MJ, Kraemer WJ, Bibus DM, Fernandez ML, Feinman RD. Carbohydrate restriction has a more favorable impact on the metabolic syndrome than a low-fat diet. Lipids 2009; 44: 297–309.
[xix]. Feinman RD, Pogozelski WK, Astrup A, Bernstein RK, Fine EJ, Westman EC, Accurso A, Frassetto L, Gower BA, McFarlane SI, Nielsen JV, Krarup T, Saslow L, Roth KS, Vernon MC, Volek JS, Wilshire GB, Dahlqvist A, Sundberg R, Childers A, Morrison K, Manninen AH, Dashti HM, Wood RJ, Wortman J, Worm N. Dietary carbohydrate restriction as the first approach in diabetes management: Critical review and evidence base. Nutrition 2015; 31: 1–13.
[xx]. Hallberg SJ, McKenzie AL, Williams PT, Bhanpuri NH, Peters AL, Campbell WW, Hazbun TL, Volk BM, McCarter JP, Phinney SD, Volek JS. Effectiveness and safety of a novel care model for the management of type 2 diabetes at 1 year: an open-label, non-randomized, controlled study. Diabetes Ther 2018; 9(2): 583–612.
[xxi]. Athinarayanan SJ, Hallberg SJ, McKenzie AL, Lechner K, King S, McCarter JP, Volek JS, Phinney SD, Krauss RM. Impact of a 2-year trial of nutritional ketosis on indices of cardiovascular disease risk in patients with type 2 diabetes. Cardiovasc Diabetol 2020; 19(1): 208.
[xxii]. Lennerz BS, Barton A, Bernstein RK, Dikeman RD, Diulus C, Hallberg S, Rhodes ET, Ebbeling CB, Westman EC, Yancy WS Jr, Ludwig DS. Management of type 1 diabetes with a very low-carbohydrate diet. Pediatrics 2018; 141(6): e20173349.
[xxiii]. Lennerz BS, Koutnik AP, Azova S, Wolfsdorf JI, Ludwig DS. Carbohydrate restriction for diabetes: rediscovering centuries-old wisdom. J Clin Invest 2021; 131(1): e142246.
[xxiv]. Evert AB, Dennison M, Gardner CD, Garvey WT, Lau KHK, MacLeod J, Mitri J, Pereira RF, Rawlings K, Robinson S, Saslow L, Uelmen S, Urbanski PB, Yancy WS Jr. Nutrition therapy for adults with diabetes or prediabetes: A consensus report.” Diabetes Care 2019; 42(5): 731–54.
[xxv]. Volek JS, Yancy WS Jr, Gower BA, Phinney SD, Slavin J, Koutnik AP, Hurn M, Spinner J, Cucuzzella M, Hecht FM. Expert consensus on nutrition and lower-carbohydrate diets: An evidence- and equity-based approach to dietary guidance. Front Nutr 2024; 11: 1376098.
[xxvi]. Institute of Medicine. Dietary Reference Intakes: The Essential Guide to Nutrient Requirements. Washington, DC: The National Academies Press. 2006. Available at: https://doi.org/10.17226/11537.
[xxvii]. NIH. Choline. Health Professional Fact Sheet. National Institutes of Health. Available at: https://ods.od.nih.gov/factsheets/Choline-HealthProfessional/